# In Silico Selection and Evaluation of Pugnins with Antibacterial and Anticancer Activity Using Skin Transcriptome of Treefrog (*Boana pugnax*)

**DOI:** 10.3390/pharmaceutics13040578

**Published:** 2021-04-18

**Authors:** Yamil Liscano, Laura Medina, Jose Oñate-Garzón, Fanny Gúzman, Monica Pickholz, Jean Paul Delgado

**Affiliations:** 1Grupo de Investigación en Química y Biotecnología (QUIBIO), Facultad de Ciencias Básicas, Universidad Santiago de Cali, Calle 5 N° 62-00, Cali 760035, Colombia; jose.onate00@usc.edu.co; 2Grupo Genética, Regeneración y Cáncer, Facultad de Ciencias Exactas y Naturales, Instituto de Biología, Universidad de Antioquia, Medellín 050010, Colombia; lauramamedina@gmail.com; 3Núcleo de Biotecnología Curauma, Pontificia Universidad Católica de Valparaíso, 2374631 Av. Universidad, Curauma 330, Chile; fanny.guzman@pucv.cl; 4Departamento de Física, Facultad de Ciencias Exactas y Naturales, Universidad de Buenos Aires and IFIBA, CONICET-UBA, Ciudad Universitaria, Pabellón 1, Buenos Aires 1428, Argentina; mpickholz@df.ub.ar

**Keywords:** peptide, antimicrobial, anticancer, *Boana pugnax*, pugnin

## Abstract

In order to combat bacterial and cancer resistance, we identified peptides (pugnins) with dual antibacterial l–anticancer activity from the *Boana pugnax* (*B. pugnax*) skin transcriptome through in silico analysis. Pugnins A and B were selected owing to their high similarity to the DS4.3 peptide, which served as a template for their alignment to the *B. pugnax* transcriptome, as well as their function as part of a voltage-dependent potassium channel protein. The secondary peptide structure stability in aqueous medium was evaluated as well, and after interaction with the *Escherichia coli* (*E. coli*) membrane model using molecular dynamics. These pugnins were synthesized via solid-phase synthesis strategy and verified by Reverse phase high-performance liquid chromatography (RP-HPLC) and mass spectrometry. Subsequently, their alpha-helix structure was determined by circular dichroism, after which antibacterial tests were then performed to evaluate their antimicrobial activity. Cytotoxicity tests against cancer cells also showed selectivity of pugnin A toward breast cancer (MFC7) cells, and pugnin B toward prostate cancer (PC3) cells. Alternatively, flow cytometry revealed necrotic cell damage with a major cytotoxic effect on human keratinocytes (HaCaT) control cells. Therefore, the pugnins found in the transcriptome of *B. pugnax* present dual antibacterial–anticancer activity with reduced selectivity to normal eukaryotic cells.

## 1. Introduction

Recently, the recorded cases of people dying from infectious diseases are on the rise owing to antibiotic-resistant bacterial strains [1]. Similarly, cancer treatments present cellular resistances and poor selectivities, thereby generating many adverse effects that result in this disease being the second most common cause of death worldwide, mainly by lung, prostate, and breast cancer [2,3]. Additionally, the resistance and low selectivity of anticancer drugs led to an increase in the pace and scope of research aimed at developing new drugs of both types [4]. Among the new molecules being researched to treat cancer and infections are peptides, characterized mainly by reduced adverse effects that evoke resistance in bacteria less often than conventional antibiotics and may also have multiple activities, such as bactericidal, hemolytic, antifungal, antiviral, healing, immunomodulating, and destroying tumor cells [5,6,7]. The relevance of bioprospecting these molecules, specifically peptides with combined antibacterial–anticancer activity (ABC), is focused on the skin of amphibians that are widely distributed throughout the planet, adapting to a variety of climates, terrains, and predators [8,9,10]. The origin of these peptides is linked to the innate immune system that dictates the speed at which pathogens are eliminated, as well as the process of angiogenesis, in wound healing [11].

One of the most frequent sources of finding this kind of peptides is the tree frog, since they have in the dorsal region of their skin, granular glands, which produce a wide variety of molecules, including ABC peptides [11,12,13]. The tree frog *Boana pugnax* is a small part of the diverse fauna that exists inside the borders of Colombia. Currently, its skin transcriptome dataset is available in the transcriptome shotgun assembly [14]. Colombia is the second most biodiverse country globally and is home to at least 51,330 described species, including the second most diverse set of anurans in the world, of which there are currently 686 reported species [15]. A substantial fraction of these species is found in the department of Antioquia, which has an area of 62,150 km^2^. According to data from the Herpetology Museum of the University of Antioquia (MHUA), approximately 30% of Colombia’s amphibian species are present in this territory, represented by 230 species (~34% of Colombia’s species) [16]. The nocturnal tree frog *B. pugnax* (Schmidt, 1857), which ranges in size from 68.9 to 77.9 mm, lives in the open lowlands of Southern Central and Northern South America [17]. Recent studies also suggested that the immune response may be incredibly efficient in frogs of the genus *Boana sp*., highlighting the potential of this genus in discovering new ABC peptides from the Colombian herpetofauna [18].

The ABC peptides could be a solution to the need to find new drugs to combat the resistance toward conventional antibiotics and anticancer drugs [7,19]. Hence, they present other characteristics that give advantages to its use, mainly: reduced adverse effects, potent activity, and broad-spectrum activity like the antibacterial–anticancer effect [20]. Some of these peptides enter the cancer cells through receptors by crossing the membranes to the interior, thereby inhibiting or activating gene expression, and affecting cancer cell homeostasis, resulting in apoptosis [21,22]. The use of antibacterial cationic peptides could also favor the destruction of cancer cells, since these cells expose anionic phospholipids on the outer leaflet of the membrane-like bacterial cells. The anionic phospholipid known as phosphatidylserine is present in smaller amounts in normal eukaryotic cells (mainly constituted by zwiterionic phospholipids) than in cancer cells. The electrostatic interaction between a cationic peptide and the anionic membrane of the cancer cell conduces to strong changes in lateral distribution and domain formation of lipids.

It was found that ABC peptides with a R/KXXR/K motif can enter the cell without causing damage to the membrane through receptors, such as neuropilin, enabling them to reach target molecules such as DNA and block the expression of genes vital to the proliferation of cancer cells [23]. However, one of the big problems with ABC peptides is cytotoxicity in normal eukaryotic cells [24]. Therefore, physicochemical properties, such as net charge, hydrophobicity percentage, hydrophobic momentum, and aggregation, become indicator tools for the design of peptides with potent ABC activity and low cytotoxicity against eukaryotic cells [7,25,26].

With computer software as support for experimental research, it was possible to reduce cost, time, and adverse effects of peptides [25,26]. Nevertheless, the physicochemical characteristics of the ABC peptides are yet to be completely elucidated, resulting in issues on how the frog skin transcriptome should be searched to identify candidates for chemical synthesis [20,21,27]. For this reason, this research used the skin transcriptome of *B. pugnax* to find and select ABC peptides through in silico methods (sequence alignment, molecular modeling, and molecular dynamics, among others). Subsequently, candidates were synthesized, and their activities evaluated against bacterial cells, human cancer cells, and normal eukaryote cells.

## 2. Materials and Methods

### 2.1. Bioinformatic Analysis

#### 2.1.1. Transcriptome

Four frogs of the species *Boana pugnax* were collected in Vegas de la Clara, Antioquia, Colombia. The frogs were anesthetized with an intercardial injection of 200 mg/kg tricaine. Skin was removed from the dorsal and ventral region under sterile conditions. Samples were homogenized, and RNA was extracted as indicated in the manuscript of Liscano et al., 2020 [14]. The transcriptome of *Boana pugnax* was assembled de novo with Trinity [28]. The transcriptome of *B. pugnax* was deposited in GenBank with the accession number Transcriptome Shotgun Assembly: data identification number: GINY010000001 (https://www.ncbi.nlm.nih.gov/Traces/wgs/GHME01?val=GINY01_accs, accessed on 12 April 2019) [14]. The coding regions were predicted based on the identification of the longest Open Reading Frame (ORF) using TransDecoder-v3 (https://github.com/TransDecoder/TransDecoder/wiki, accessed on 12 April 2019).

#### 2.1.2. Databases

Five cured peptide databases, AVPdb (Antiviral database), with 2683 antiviral peptides were used for subsequent transcriptome alignments [29]; APD3 (The Antimicrobial Peptide Database) with 3136 peptides [30], Cancer ppd with 3491 peptides [31], Cell-penetrating peptides (Cpp) with 1700 peptides [32], and Signal peptide database with the 13,094 mammalian peptides [33].

The article by Xu and Lai, 2015 was also used [13] with 2000 peptides. The authors’ healing peptides were also used like Chung et al., 2017, Mu et al., 2014, Mangoni et al., 2016, Tang et al., 2014 and Xiao et al., 2016 [34,35,36,37,38].

#### 2.1.3. Alignments from the Databases

FASTA software version 36 was used [39] for alignments between databases and the transcriptome. The algorithm used was glsearch36l this allows to compare proteins-proteins [39]. Among the parameters used are the E-value 1 × 10^−3^, and the one to show the five best alignments with the best Score. The E-value threshold chosen was not demanding compared to a 1 × 10^−5^ threshold; that is, the lower the higher E-value is the probability that the alignment has a high similarity between the two sequences, that of the database and the one obtained from the transcriptome [40,41]. This was done because the first part of the search was exploratory; that is, there was no indication of how many peptides would be obtained from the transcriptome. In subsequent analyses, the selection of these sequences was refined.

#### 2.1.4. Obtaining Physicochemical Parameters of Peptides

Information on its physicochemical properties, such as net charge, percentage of hydrophobicity, hydrophobic momentum, and sequence length, was obtained from the peptide sequences found in the transcriptome. The sequences of the peptides were placed in the software online, like the Thermo Fisher Peptide Analyzing Tool (https://www.thermofisher.com/co/en/home/life-science/protein-biology/peptides-proteins/custom-peptide-synthesis-services/peptide-analyzing-tool.html, accessed on 12 April 2019), HELIQUEST (https://heliquest.ipmc.cnrs.fr/, accessed on 12 April 2019), and Pep-calculator (https://www.pep-calc.com/, accessed on 12 April 2019) [42,43,44,45]. The secondary structure was predicted with NP: network protein sequence analysis (https://npsa-prabi.ibcp.fr/cgi-bin/npsa_automat.pl?page=/NPSA/npsa_server.html, accessed on 12 April 2019) using the MLRC method [46,47]. The last parameter determined was the tendency to aggregate the peptide in aqueous solution, using the TANGO software [48].

#### 2.1.5. In Silico Prediction of Antimicrobial Activity, Cell-Penetrating Peptide and Anticancer Using Support Vector Machine

For the predictions of the activities, the statistical learning method of Support Vector Machine (SVM) was used, which presents a set of supervised classification and regression algorithms that use various characteristics of the peptides, such as amino acid composition, dipeptide composition, and physicochemical parameters to build prediction models of antimicrobial, cell-penetrating peptide, and anticancer activities [34,49,50,51].

To predict the antimicrobial activity, Collection of Anti-microbial Peptides (CAMP) prediction SVM was used; for cell-penetrating peptidem Cell penetrating peptide (Cellpp) SVM and Anticancer peptide (Anticp) SVM were used for anticancer prediction. The values are between 0 and 1, which indicates that between closer to 0, it is less likely to have the activity, and the closer to 1, the peptide is more likely to possess the activity [33,34,52].

#### 2.1.6. Filter for the Selection of Candidates with High Probability of Presenting Combined Antibacterial-Anticancer Activity (ABC) and Being Cationic with Helical Structure

From the results obtained from the previous alignment, 904 peptides were obtained, of which it was expected to achieve at least four candidates with the highest probability of being cationic helical with dual antibacterial–anticancer activity (ABC). The first filter was the elimination of duplicate sequences and was followed by the selection of peptides with the E-value < 1 × 10^−4^.

With the purpose of finding sequences not reported by the literature, patents, and databases, those that had an identity between 50% and 95% with a similarity >90% were selected in order to have sequences with greater probability of having the same structure and activity of the peptide from the database [47]. In order to improve the selectivity in terms of the desired activities, probabilities of being an antimicrobial peptide with CAMP prediction SVM greater than 0.90 were sought, and, similarly, the cell-penetrating prediction SVM and Anticancercp greater than 0.60 were used.

Added to these filters were attached two final filters, the net charge between +4 and +9 so that they had a powerful antibacterial activity and little hemolysis [48,49,50]. The last filter was the helix structure, since peptides with ABC activity have this structure linked to membrane stability and antimicrobial potency [21,51].

#### 2.1.7. Phylogenetic Analysis and Search for De Novo Motifs in Candidate Sequences after the Filter

A phylogenetic analysis was performed, whose objective was the identification of the relationship between the four candidate sequences found in the transcriptome in relation to the peptide with which it was hit in the cell-penetrating peptide database called DS4.3. This analysis was performed with the MEGA software version 10 [52]. The method to build the phylogenetic tree was the Neighbor-joining [53]. An inferred bootstrap consensus tree of 1000 repetitions was performed [54].

For the discovery of de novo motifs among the five sequences, the online tool MEMEsuite (http://meme-suite.org/, accessed on 12 April 2019) was used. The identification of these reasons is because many of them are related to biological functions, and their detection is important for studies of molecular interactions in the cell [55].

#### 2.1.8. Molecular Modeling of Candidate ABC Peptides

Peptide models were obtained using I-TASSER (https://zhanglab.ccmb.med.umich.edu/I-TASSER/, accessed on 2 April 2019) as a tool to align the peptide sequence with the database of the protein bank, RSCB protein data bank (https://www.rcsb.org/, accessed on 12 April 2019). The model proteins with the best alignment to the peptide sequences were chosen to represent the 3D structure. From 2R9R models (https://www.rcsb.org/structure/2R9R, accessed on 12 April 2019) [56] and 4BGN (https://www.rcsb.org/structure/4bgn, accessed on 12 April 2019) [57] found by I-TASSER, 100 molecular models were constructed for each peptide using MODELLER version 9.14 (https://salilab.org/modeller/, accessed on 12 April 2019) [58]. The models were built using the default auto-mode methods and environmental classes of MODELLER. The final models were selected according to the discrete optimized protein energy score (DOPE score). This score evaluates the energy of the models and indicates the best probable structures. The best models were evaluated through PROSA II (https://prosa.services.came.sbg.ac.at/prosa.php, accessed on 12 April 2019) and Molprobity (http://molprobity.biochem.duke.edu/, accessed on 12 April 2019) [59,60,61]. This verifies the stereochemical quality of a protein structure through the Ramachandran graph, where good-quality models are expected to have >90% amino acid residues in the most favored and additional regions allowed, while PROSA II indicates the quality of the fold. The structure visualization was done in PyMOL (http://www.pymol.org, accessed on 12 April 2019).

### 2.2. Molecular Dynamics

#### 2.2.1. Water Box System Construction

A 100 Å box (Armstrongs) was created in the bucket with water molecules and KCl at a concentration of 0.1 M with the online software CHARMM-GUI [62]. The pugnins and the original DS4.3 peptide were located in the center of each box, generating a total of three systems. The systems used a temperature of 310° K and ran for ten nanoseconds (ns).

#### 2.2.2. Gram-Negative and Gram-Positive Bacterial Membrane Models Construction

A basic membrane system (phospholipids only) of *E. coli* (Gram-negative) was created with the CHARMM-GUI online software [62]. The *E. coli* membrane according to Epand et al., 2009 has two types of POPE phospholipids (1-palmitoyl-2-oleoyl-sn-glycerol-3-phosphatidylethanolamine) and POPG (1-palmitoyl-2-oleyl-sn-glycerol-3-[phospho-rac]-(1-glycerol)]), distributing 80 POPE and 20 POPG molecules in the outer and inner monolayers [63]. The *S. aureus* membrane was constructed with cardiolipin (PMLC1) and POPG, distributing 60 POPG and 40 PMLC1 molecules in the outer and inner monolayers [64].

#### 2.2.3. Molecular Dynamics of Candidate ABC Peptides

Molecular dynamics simulations were performed with NAMD software version 2.13 [65]. The concentration of KCl was 0.15 M, as determined in the Ion Placing Method, with a water thickness of 22.5 Å and using the forcefield CHARMM36m as a force field [66]. Systems were adjusted slowly by heating at a temperature of 310 ° K at 1 fs (femtosecond)/step for 75 ps (picoseconds) to the conformation of the peptide and is completely inserted into the membrane to ensure that the system has no steric clashes or inappropriate geometry, relaxing the structure in a process called energy minimization and 300 ps at 2 fs/step for the equilibrium phase.

Once the system is balanced at the desired temperature and pressure: We execute the molecular dynamics (MD) of production for data collection for 10 ns. For molecular dynamics, an unbound limit of 12 Å was used simulating for 10 nanoseconds. The sum of Particle Mesh Ewald (PME) was applied to correct long-range electrostatic interactions [67]. The bonds containing hydrogen atoms involved were restricted using the SHAKE algorithm to their minimized energy values [68], which allowed a numerical integration time step of 2 fs to be used in the simulation [69]. The helical structure has structural importance for antibacterial activity; therefore, to determine the stability of this structure during the 10 nanoseconds ns or 10,000 ps, Visual Molecular Dynamics (VMD) were used [70] and GROMACS [68]. In the VMD analysis, the secondary structure during the simulation time was determined using the STRIDE algorithm [71]. To work with GROMACS (Version 2019.3), the .dcd format generated by NAMD were converted to .xtc using MDTRAJ [72], with the “mdconvert” command. Subsequently, the root mean squared deviation (RMSD) of the backbone of the peptide (backbone) and the RMSF (root mean squared Fluctuation) per residue were obtained using the gmx rms and gmx rmsf commands. For the flexibility analysis, the ∆RMSF (nm) was made, making the difference between the RMSF of the pugnin and the RMSF of the original DS4.3 peptide. The interaction analysis was performed with the Discovery studio visualizer software (http://accelrys.com, accessed on 12 April 2019) to find the amino acids that interact with the molecules.

### 2.3. Synthesis, Characterization and Circular Dichroism of Peptides

Peptides sequences were synthesized using a Liberty Blue automated microwave peptide synthesizer (CEM Corp., Matthews, NC, USA) following a standard Fmoc/tBu, purification, characterization and circular dichroism protocols by Luna et al., 2016 [73].

### 2.4. Antimicrobial Test

The antimicrobial test was performed according to the CLSI protocol (plate microdilution technique) [74] with modification in the reading form (absorbance) and the medium used, because this was done by means of a Thermo Multiskan GO brand spectrophotometry microplate reader using the TSB medium (tryptic soy broth) and calculating the percentage of growth and inhibition of the strain with the absorbances according to the following Formulas (1) and (2):%Growth = peptide absorbance (peptide + strain + medium) − Blank abs (Medium) × 100/Abs negative control (medium + strain) − Blank abs (Medium)(1)
%Inhibition = 100 − %Growth(2)

Pure bacterial cultures were incubated at 37 °C for 18 to 20 h. A colony from the pure culture was resuspended in sterile water to reach the turbidity of 0.5 McFarland, and the resulting suspension contained 1–4 × 10^8^ colony forming units (CFU/mL). The minimum inhibitory concentration 50 (MIC50) was calculated using lineal regression [75]. The MIC50 is the lowest concentration of drug that reduces the bacterial population by more than 50% [76]. MBC (minimum bactericidal concentration, 100% growth inhibition) was calculated by plating out the contents of the first three wells showing no visible bacterial growth onto TSB agar plates incubated at 37 °C for 18 to 20 h [77]. The strains used were *Staphylococcus aureus ATCC 25923, Pseudomonas aeruginosa ATCC 27853, Escherichia coli ATCC 25922,* and *Enterococcus faecalis ATCC 29212.* The peptides evaluated were pugnin A and B at the concentrations of 100 uM, 33.33 µM, 11.11 µM, 3.7 µM y 1.23 µM.

### 2.5. Hemolytic Test

Erythrocytes were isolated from fresh human peripheral blood. They were washed three times with PBS by centrifugation for 5 min at 800× *g* and resuspended in PBS. Peptides in different concentrations (100, 33.33, 11.11, 3.7, and 1.23 µM) were added to 4% human erythrocytes in PBS and incubated at 37 ° C for 1 h. Then, the mixtures were centrifuged at 800× *g* for 5 min. Aliquots of the supernatant were transferred to 96-well plates. Hemolysis was measured by absorbance at 540 nm with a Multiskan Go (Thermofisher, Waltham, MA, USA). For negative and positive controls, erythrocytes were used in PBS and erythrocytes in PBS with 1% Triton X-100, respectively. The percentage of hemolysis was calculated using the following equation: [(Abs in peptide solution − Abs in PBS)/(Abs Triton x-100 − Abs in PBS)] × 100. The hemolytic concentration 50 (HC50) was calculated, which is the necessary concentration of peptide to induce 50% of the lysis of erythrocytes under physiological conditions.

### 2.6. Cell Lines

In vitro biological tests were performed on three cell lines; the HaCaT (Human Keratinocytes Cells) line was the control of normal eukaryotic cells to determine whether or not there was damage from peptides. Then, two cancer cell lines were evaluated to determine the cytotoxic effect of the peptides on them. One of these was the human prostate cancer cell line (PC3, ATCC Number: CRL-1435) and the other was the breast cancer cell line (MFC7, ATCC Number: HTB22). The HaCaT cells were cultured in DMEM medium, 5% fetal bovine serum, 100 U/mL penicillin and 100 mg/mL streptomycin, and incubated at 37 °C in a humid atmosphere and 5% CO_2_. MCF7 and PC3 cells were cultured in monolayer in T25 cell culture bottles, in DMEM medium. The medium was supplemented with 5% fetal bovine serum (FBS Gibco), 100 U/mL penicillin and 100 ug/mL streptomycin (Gibco) and incubated at 37 °C, with CO_2_.

#### Cytotoxicity Test (MTT) 3-(4,5-dimethylthiazol-2-yl)-2,5-diphenyltetrazolium Bromide

The cells used for the test were HaCaT, PC3, and MFC7. The cells were seeded in a 96-well plate at a density of 10,000 cells/well and cultured with the pugnin peptides A and B, at the concentrations of 100 µM, 33.33 µM, 11.11 µM, 3, 7 µM, and 1.23 µM until 24 h. Then, 10 μL of MTT solution (5 mg/mL, Sigma) were added to each well and incubated at 37 °C in 5% CO_2_ for 4 h. After centrifugation at 3000× *g* for 15 min, the supernatant was removed, and acid isopropanol (dimethylsulfoxide, Sigma) was added in the volume of 100 µL to dissolve the formazan crystals. The absorbance was measured at 570 nm using a microplate reader (Varioscan Flash, Thermo). For the negative and positive controls, erythrocytes in PBS and erythrocytes with Triton X-100 at 1%, respectively, were used.

Then, with the absorbances obtained, the calculation of the percentage of cytotoxicity of the peptides with respect to each of the cell lines was performed, using the following Formulas (3) and (4):%Citotoxicity = 100 − %Viability(3)
%Viability = (peptide absorbance − blank absorbance)/(Non treated cells absorbance − blank absorbance) × 100(4)

### 2.7. Flow Cytometry

#### 2.7.1. Evaluation of Mitochondrial Membrane Potential and Cytoplasmic Membrane Integrity

To determine variations in mitochondrial membrane potential and cytoplasmic membrane damage, HaCaT and PC3 cells were analyzed using DiOC_6_ (Molecular Probes D273) and propidium iodide (PI) (Sigma P4170). The cells were suspended in 500 μL of PBS containing 5 μL of 10 μM DiOC_6_ and 5 μL of 1 mg/mL of PI, and then the cell suspensions were incubated in the dark at room temperature for 20 min, washed, resuspended in PBS, and analyzed by flow cytometry (BD LSR Fortessa). The concentrations of pugnin A and B peptides were 50, 100, and 150 μM, with three repetitions [78].

#### 2.7.2. Apoptosis Analysis

To perform this test, the Annexin V/Dead Cell Apoptosis Kit with SYTOX (Invitrogen, Carlsbad, CA, USA) apoptosis detection kit was used. This kit is based on the observation that from the initial stages of apoptosis, in most cell types, translocation of the phosphatidyl serine (PS) membrane phospholipid occurs, from the inner face of the plasma membrane to the outer surface [79]. Once on the cell surface, PS can be easily detected by staining with a fluorescent conjugate of Annexin V, a protein that has a strong natural affinity for PS. The set of Annexin V-PE with Sytox was used, which is a vital dye. The cells were seeded at a density of 1.5 × 10^5^ cells per well in 24-well plates and then treated with the peptides. After a treatment time of 24 h, the cells were washed with PBS and resuspended in 200 µL of 1× Buffer Binding; 10 μL of Annexin V-PE and 10 μL of SYTOX were subsequently added. Cytometry tubes were vortexed gently and incubated for 20 min in the dark at room temperature. Two independent biological replicates were performed in this assay for each cell line-peptide-concentration set.

#### 2.7.3. Cell Cycle Analyses

Cell samples were fixed in 70% ethanol and subsequently incubated with 100 μg/mL RNase (Sigma, St. Louis, MO, USA, R5000), stained with 100 μg/mL propidium iodide (Sigma, P4170) for 30 min, and analyzed for PI fluorescence using an EPICS XL flow cytometer. Percentages of cells in each phase were calculated using FlowJo version 10 (https://www.flowjo.com/, accessed on 12 April 2019). Three independent experiments were carried out for each cell line. The data are presented as mean ± SEM. 

### 2.8. Statistics

The experimental design for the antimicrobial and hemolytic tests consisted of three repetitions per sample and three replicates that were statistically analyzed by the two-way ANOVA to determine the interaction between factors and the influential variable with a significance level of 0.05. “The bidirectional ANOVA comes with a lower probability of Type II error (that is, greater statistical power) because the three contrast tests maximize the sample size for each test” [80]. For the statistical analysis of the MTT cytotoxicity test, the one-way ANOVA was used with post-hoc comparisons performed by the Fisher program with minimum significant difference tests (FPLSD), using three repetitions per sample and three replicates with a level of significance of 0.05 [81]. A *p* value < 0.05 was considered statistically significant [82,83]. The analysis of variance (ANOVA) was performed through the software Prism trial version 8 (https://www.graphpad.com/scientific-software/prism/, accessed on 12 April 2019).

## 3. Results

Figure 1 shows the whole process from the alignment of the peptide databases using the transcriptome of the skin of the frog *B. pugnax* until the final pugnins A and B. Peptides with 100% identity were found, representing already reported peptides by the literature in other species. Therefore, we searched for new peptide sequences in the transcriptome, obtaining 375 putative sequences. In order to find peptides with dual antibacterial-anticancer activity, a filter including physicochemical parameters and artificial intelligence algorithms was performed, which are described in detail below. Therefore, four candidate putative sequences derived from the cell penetrating peptide DS4.3 were obtained. Two pugnins A and B were synthesized and subjected to antibacterial analysis, cytotoxicity tests, circular dichroism, and flow cytometry in order to corroborate the in silico prediction.

### 3.1. Searching ABC Peptides from B. pugnax Transcriptome

#### 3.1.1. Alignment Transcriptome to Peptides Databases

As a first step for obtaining the pugnins, an alignment was made between the *B. pugnax* transcriptome and seven peptide databases, as mentioned in Figure 2. In this way, 375 peptide sequences were obtained without repetitions from the transcriptome, ranging from those with 100% identity, indicating the presence of peptides already reported, and others with lower percentages of identity, leading to the possibility of new putative peptides without confirmed activities that may possess the wanted antibacterial or anticancer activities.

From database alignments with the *B. pugnax* transcriptome, 166 putative sequences were obtained from the Xu et al., 2015 database, 82 from the Signaling Peptide database, 76 from APD3, 27 from the Cell Penetrating Peptide database, 15 from AVP Antiviral, 5 from Cancerppd, and four other putative peptides from the wound healing peptide manuscripts reported by Mu et al., 2014, Tang et al., 2014, Mangoni et al., 2016, Xiao et al., 2016, and Chung et al., 2017 [36,37,39,40] (Figure 2).

From the 375 putative peptides found in the transcriptome, selection of the peptides that had 100% identity was carried out, meaning that the sequences found in the transcriptome was of high similarity to the database presenting the structure, activity, and physicochemical parameters, and equal to those of the original peptide (database peptide used for transcriptome alignment). In this case, 20 peptides with 100% identity were identified (Figure 3).

Out of these 20 peptides, five of them possessed distinct myotropic and antiviral properties. Among the myotropic peptides, tryptophyllin, fallaxidin, dentatin, demorphin, and HATL-5 were identified. They were referred as peptides with myotropic activity because they exerted contractile effects on smooth muscles [13]. In addition to this activity, tryptophyllin was also found to possess liver protein synthesis and bodyweight activity, and is believed to have insulin-releasing activity as well [84,85,86].

Asides from being a myotropic with a structure similar to opiates, demorphin was found to exhibit analgesic properties in rodents and primates, and exhibited antimicrobial ability in frogs: *Phyllomedusa bicolor* and *Phyllomedusa sauvagii* [87,88]. Another peptide found was dentatin, which in humans was found to exhibit inhibitory effects on breast, prostate, and liver cancer cells [89]. In these, it induced apoptosis by the accumulation of reactive oxygen species (ROS), downregulation of the expression levels of antiapoptotic molecules (Bcl-2, Bcl-xl, and Survivin), and possible inhibition of *NF-κB* [2,89].

The Buforin II peptide, which was part of the peptides with 100% identity found in the transcriptome of *B. pugnax,* was also found to possess anticancer activity. This peptide was derived from Buforin I, isolated from the stomach of *Bufo bufo garagrizans*, which also exhibited antibacterial, antifungal, and cell-penetrating activities [90]. It was characterized as an helical cationic peptide with hydrophobicity of 33% [91].

The other antibacterial peptides found were Ubiquicidine, cgUbiquitina (origin: Pacific oyster, *Crassostrea gigas*) [92], Fusaricidine C and D (originally found in *Bacillus polymyxa*) [93], Peptide 3910 (originally found in *Sus scrofa*) [94], and Histone H2A.

Antiviral peptides: AVP427 and AVP1924 were also identified, which were recorded to exhibit inhibitory effects against the respiratory syncytial virus [95]; AVP1487 against papillomavirus [96] and AVP1173 against hepatitis C virus, respectively [97].

AntiVIH HIP1102, a cationic helical peptide, also known as penetratin, and as its name implies, possesses the ability to penetrate cancer cells. Thus, its current use in anticancer therapy as carriers of anticancer molecules into cancer cells [98,99,100]. Cell-penetrating peptides such as these are also used to carry antivirals, specifically antiHIV as a strategy to combat the virus [101].

Other peptides found in the transcriptome were Angiotensin 8, the signal peptide LRC3B_HUMAN (Leucine-rich repeat-containing protein 3B), and the antioxidant APBMH. This antioxidant is used on the skin of amphibians to combat ROS that peroxidate lipids, denature proteins, and damage DNA [13,102].

Figure 3 also shows the comparison of three physicochemical parameters, the length of the sequence, its net charge, and percentage of hydrophobicity. In general, it is confirmed that these peptide lengths varied between 5 and 20 residues, with a net charge running between −5 and 6. It is also noteworthy that the variation of hydrophobicity was wide, ranging between 5% and 60%. By type of activity, myotropic peptides had a net charge of 0 with lengths of 5 and 6 aa. The antivirals were anionic, with hydrophobicity between 40% and 60%, while those with antibacterial activity were mostly cationic with varying lengths and hydrophobicities.

#### 3.1.2. Filtering the 375 Peptides to Obtain Peptide Candidates for Chemical Synthesis

The funnel diagram was used to reduce the number of peptides and obtain the candidate peptides for chemical synthesis (Figure 4). Here, 904 peptide sequences were obtained and subjected to duplicate elimination, which reduced the number of peptide candidate to 699. However, the number of sequences remained very high. Therefore, the E-value was made more restrictive, from 1 × 10^−3^ to 1 × 10^−4^, which resulted in 375 peptides. At this point, the resulting peptides were aimed at the cationic helical antibacterial–anticancer (ABC) peptides not reported in the literature.

To be considered as “new” peptide sequences, we consider that the percentage of identity must be between 50% and 90%. Other authors mention that values above 30% ensure that sequence, structure, and activity are very similar to those of the peptides used as templates [47].

Up to this point, 69 sequences were obtained. Then, the machine learning algorithms of support vector machine (SVM) were used to predict the probability of these peptides in being antimicrobial, anticancer, and cell-penetrating peptides, reducing the number of peptides to six candidates. Finally, among the latter, the positive net charge and alpha-helix formation were prioritized, resulting in three final candidates.

These three candidates are derived from the DS4.3 peptide, which is a cell-penetrating peptide derived from the S5 subunit of a voltage-dependent potassium channel of the genera *Drosophila* [103]. However, when reviewing the 375 peptides, it was observed that there were four sequences derived from DS4.3, but one was eliminated in the filter because the probability of being antimicrobial was very low according to the SVM algorithm. We decided to stick with the four predicted peptides even though one of them, pugnin B, did not have a good prediction as an antibacterial according to the SVM algorithm of CAMP (Figure 5). This choice was based on the high similarity of pugnin B with the sequence of pugnin A and with the DS4.3 peptide that obtained a high SVM value, i.e., high probability of being antibacterial.

### 3.2. Comparison of Physicochemical and Structural Characteristics between Helical Cationic Peptides with Probable Combined Antibacterial–Anticancer Activities

The four candidates were compared with the DS4.3 peptide, and given the name pugnins, which ranged from A to D because of their origin from the skin of *B. pugnax*. As shown in Figure 5, the four putative peptides had the same sequence lengths (13 aa), hydrophobicity between 39% and 46%, and a net charge of +4 for pugnin C and +5 for everyone else. Hydrophobic moments were also analyzed, where pugnin B exhibited the lowest value (0.08), while the others showed values around 0.30. SVM was also used to compare their antibacterial, anticancer, and cell-penetrating peptide activities. Concerning the antibacterial SVM, the probability was high in all pugnins except pugnin B. For this reason, the sequence was ruled out. The probability of being cell-penetrating was the same in all pugnins, while the probability of possessing anticancer activity presented an average value of 0.66, except for pugnin D with a value of one; this meant that the probability of being anticancer was high in this peptide shown below.

The sequence analysis results are as shown in Figure 6 below. It was evident that the KLAR sequence was found at the end of all the peptides, with the exception of pugnin C. Also, through the phylogenetic analysis of the sequences, it could be inferred that the most related to the DS4.3 peptide were pugnin A and B. This was because alignments with BLAST-p revealed that they came from voltage-mediated potassium channel proteins (KCNS3), just like the original peptide. Results also showed that pugnins C and D were more related to each other and belonged to calcium channel proteins (Figure 6B).

The common motif found between the pugnins and the DS4.3 peptide was xRxxRxxK/RxxR (X = hydrophobic residue, R = arginine, K = lysine). This motif demonstrated the importance of hydrophilic amino acids, such as arginine and lysine, for every two hydrophobic residues. The presence of the R/KXXR/K motif in both the amino-terminal and the carboxyl-terminal of the peptides was highlighted (Figure 6C). The R/KXXR/K motif (arginine/lysine, two hydrophobic amino acids, arginine/lysine) interacts with neuropilin 1 or 2 receptors of cancer cells, and then enter by endocytosis, thus increasing the efficiency of these peptides to cross tumor barriers [23].

Aggregation between the peptides was also evaluated using the TANGO online software, which presented aggregation with only pugnin B. Figure 6D shows the residue aggregation score. The IFWVI sub-sequence of pugnin B also had a 75% probability of aggregation, which would present a difficulty, since the peptides that were added failed to interact properly with the membrane or receiver to carry out its function [104].

### 3.3. Pugnins Modeling and Molecular Dynamics Analysis

Pugnin A and B were chosen for molecular modeling, molecular dynamics, and in vitro assays, because they had strong similarities to the peptide DS4.3, which is a cell-penetrating peptide. These two pugnins showed a high probability of penetrating cancer cells.

The structures obtained from 3D modeling revealed that both the pugnins and the DS4.3 peptide had secondary alpha-helix structure. These structures were validated with PROSA and Molprobity. PROSA was used to observe the Z-score inside the region of these structures obtained by nuclear magnetic resonance, to obtain a high fidelity of the modeling. Molprobity results showed that 100% of the waste was obtained in the favorable region of the alpha-helix formation, indicating that our 3D models have the appropriate parameters for a protein structure [105].

With the recent developments in high computational performance, the use of molecular dynamics (MD) simulations increased as a support tool for analyzing behaviors of molecules at scales that were previously impossible to study [70]. In MD simulations, Newton equations were numerically solved, given as a result of the trajectories that could be further analyzed. Tools, such as RMSD and RMSF, allow for the exploration of the conformational changes of a given molecular system [106].

Figure 7 shows the RMSD analyzes of the pugnins (A, B) and the DS4.3 peptide. Figure 7A compares the DS4.3 peptide and pugnins A and B. Here, it was revealed that the six nanoseconds (ns) structures showed variations in the RMSD, the DS4.3 achieving a great variation in its structure of an approximate value of 0.3 nm. Pugnin B on the other hand had minor variations in the order of 0.2 nm. These results were corroborated with the notorious structure changes in the DS4.3 peptides and pugnin A shown in Figure 8A. Helix stability was lost at the amino and carboxyl ends of pugnins A and B, being conserved only in the region comprising the IFW residues.

Figure 8 shows how pugnins A and B lost their helical structure in water, presenting themselves as random coil while in the *E. coli* membrane this structure is maintained except for the amino terminal ends, where we find arginines that form hydrogen bonds with water. Therefore, pugnins A and B in water form random coil, and when they come in contact with the membrane, they adopt the helical structure [107].

Variations in helicoidal structure were found more in the amino-terminal region, where the first two or three residues (arginine’s presence) of the peptides lost the alpha helix. This loss of stability of helicoidal structure was most noticeable in pugnin B. These changes in stability could be reflected in the antibacterial activity, most often related to the decrease in antibacterial potency (see Figure 8) [108].

The ∆RMSF analysis in the *E. coli* membrane model was done to compare the residues’ flexibility throughout the simulation [109] between the pugnins, as observed in Figure 9. The greatest flexibility was found with pugnin A, with the first two residues R, L, and M of the amino-terminal region, having a maximum ∆RMSF value of 0.04 nm. The greater flexibility of pugnin B was almost the same for MRIFW residues in the central region and arginine of the carboxyl-terminal region with a value of approximately 0.01 nm.

With these ∆RMSF results, it is confirmed that the regions of greatest variability in the pugnins during the simulation in the *E. coli* membrane model were at the amino and carboxyl-terminal ends, where the arginine residue associated with a hydrophobic residue is found.

Figure 10 illustrates the behavior of the pugnins in the *S. aureus* membrane model system. Pugnin A shows a greater variation in its structure at 2, 5, 6, and 8 ns of 0.06 nm with respect to the greater variation of pugnin B, which was 0.04 nm at 8 ns. Both pugnins from 8 ns to 10 ns the variation in their structure decreases.

Figure 11 shows the intermolecular interactions between the pugnins and the components of the *S. aureus* membrane model system during the 10 ns. Both pugnins were dominated by hydrogen bonds mainly with water molecules. The hydrophobic interactions were maintained between 20 and 25 mainly with pugnin B. Pugnin A showed a slightly higher number of interactions with POPG than pugnin B during the 10 ns. In contrast, pugnin B presented a slightly higher number of interactions with PMLC1 with respect to pugnin A during the 10 ns.

The residues pugnin A, arginine (ARG1, ARG4, ARG7, ARG13), and lysine (LYS10) had the highest number of hydrogen-bonding type interactions, mainly with water molecules as seen in Figure 12A. These same residues were important in interacting with membrane phospholipids through hydrophobic interactions. However, the pugnin A residues that established the highest number of hydrophobic interactions were ILE8, PHE6, LEU9, and ALA12. In pugnin B, again arginines (ARG1, ARG4, ARG13) and lysine (LYS10) presented the highest number of hydrogen bonding type interactions. For pugnin B, the hydrophobic interactions interacting with phospholipids were with residues MET2, MET3, PHE6, ALA12, and TRP7 with the highest number of this type of interactions.

### 3.4. RP- HPLC Chromatography, Mass Spectromety, and Circular Dichroism

After the solid-phase synthesis of pugnins A and B was conducted, the peptides were purified on columns C-18. Pugnin A was eluted using 30% acetonitrile, while pugnin B was eluted using 20% acetonitrile. This is because hydrophilic molecules elute with a high percentage in water, while hydrophobic ones elute with high percentage of acetonitrile [110]; therefore, pugnin A could be said to be more hydrophobic than pugnin B.

Once the purifications were done, the quality of the peptides were verified using reverse phase chromatography (RP-HPLC) and mass spectrometry (MS). The results obtained for the three pugnin peptides by RP-HPLC, with a gradient of 0–70% Acetonitrile (ACN) in 20 min, with a retention time of 13,289 min for pugnin A and 12,852 min for pugnin B, were recorded (see Appendix A). In the chromatograms, a defined peak was observed in the retention times mentioned, which was an indicator of high purity of each peptide.

In relation to the results of characterization by MS, the values of the theoretical ions showed high similarity with the theoretical values calculated for each peptide, with variations less than 0.01% (see Appendix A). This indicated the presence of the peptides synthesized in the spectrograms. Therefore, it could be concluded that the synthesis was successful with pugnins A and B for subsequent antibacterial analysis, hemolytic test, cytotoxicity test, viability, and apoptosis of cancer cells.

Before the in-vitro analyses were performed, the secondary structure of the pugnins were confirmed through circular dichroism (Figure 13). It was discovered that in water, the structure formed in both peptides were random coils, and in 30% tetrafluoroethylene (TFE) they formed alpha helixes, this was predicted using molecular modeling and MD. Pugnins in water were also found to have low helicity, and do not form any helical structure. This form of structure is known as a random coil. This may be because of the unfavorable energy transfer of hydrogen bridges with the non-helical peptide skeleton in the water [111]. Intrapeptide hydrogen bridges play a fundamental role in folding and forming the alpha helix, which are favored in amphipathic environments, such as membranes, and the distribution of hydrophilic and hydrophobic residues in peptides affect the interaction with membrane phospholipids [112].

### 3.5. Antibacterial Test

The antibacterial test was performed using gram-positive (*S. aureus*, *E. faecalis*) and gram-negative (*E. coli*, *P. aeruginosa*) bacteria; with each strain, peptide concentrations ranged from 1.23, 3.7, 11.11, 33.33, and 100 µM. These concentrations were chosen by a previous screening test with a concentration of 30 µM, identifying an inhibitory process by which the peptides in each of the four strains were evaluated.

Therefore, the concentration ranges of the peptides were expanded to obtain the MIC_50_, MIC_90_, and Minimum bactericide concentration (MBC). Table 1 shows the four strains evaluated, with the respective MIC_50_ (Minimum Inhibitory Concentration at 50% bacterial inhibition), MIC_90_ (Minimum Inhibitory Concentration at 90% bacterial inhibition), and MBC for each peptide. These MIC_50_, MIC_90_, and MBC were calculated from the equation of the line for each peptide, as illustrated in Appendix A and Appendix A.

Pugnins had a greater antibacterial effect on gram-negative bacteria than on gram-positive ones. Pugnin A had higher antibacterial effects in gram-positive bacteria than pugnin B, exhibiting the lowest MIC_90_ (149 µM) and MBC (183 µM) in *S. aureus* ATCC 25923, and the lowest MIC_90_ (107 µM) in *E. faecalis* ATCC 29212. The antibacterial effect of pugnin A against gram-negative bacteria was also better in *P. aeruginosa* ATCC 27853, with the lowest MIC50 (4.1 µM), MIC_90_ (28.5 µM), and MBC (34.6 µM). Alternatively, pugnin B exhibited better inhibitory effect on the growth of *E. coli* strain ATCC 25922.

### 3.6. Hemolytic Test

The results of hemolytic activity were evaluated with the hemolytic concentration (HC50). Figure 14 below shows the percentage of hemolytic activity of the pugnins, obtained through the equation of the line. The HC50 of pugnin A was 17 µM (R^2^ = 0.99), followed by pugnin B with 32.21 µM (R^2^ = 0.98). Appendix A shows the two-way ANOVA hemolytic activity of the pugnins, taking a value *p* < 0.0001 for all the evaluated models. Therefore, it could be concluded that there is a statistically significant association between the response variable (hemolytic activity) and factors (peptide concentrations).

### 3.7. MTT Cytoxicity Test

The evaluated concentrations of the peptides were 1.23, 3.7, 11.11, 33.33, and 100 µM; however, only at 100 µM was cytotoxic effect of the pugnins on the cells observed. Figure 15 shows the percentage of cytotoxicity of each of the pugnins on the mentioned cell lines. For HaCaT cells, a greater cytotoxic effect was observed with pugnin A (60% cytotoxicity) than with B (13% cytotoxicity). The cytotoxic effect on MFC7 cells, on the other hand, was greater in pugnin A (12% cytotoxicity) compared to B (5% cytotoxicity) at a concentration of 100 µM, while the cytotoxic effect on PC3 was greater in pugnin B (90% cytotoxicity) compared to A (74% cytotoxicity) at a concentration of 100 µM.

For HaCaT, MFC7, and PC-3 cell lines treated with pugnins A and B, their statistical significance was determined using a one-way ANOVA, when comparing the two pugnins at 100 µM (*p* < 0.05, 95% CI). In relation to the post-hoc comparison of the control with the treatments using the Dunnett’s test, the *p*-value results were statistically significant for the relation control vs. pugnin A and control vs. pugnin B in the HaCaT and MCF7 lines. However, for the PC3 line, the control vs. pugnin B ratio was not statistically significant (see Appendix A).

### 3.8. Evaluation of Retention of DIOC_6_ and Incorporation of Propidium Iodide by Flow Cytometry

The dual staining with DiOC_6_ combined with propidium iodide (PI) allows distinguishing live cells from dead cells by measuring the mitochondrial membrane potential and the cell membrane integrity. Live cells are DiOC_6_ positive and PI negative (low right panel), while dead cells can be on early stages of apoptosis (negative for PI with decreased DiOC_6_ fluorescence), late stage of apoptosis (DiOC_6_+/PI^+^, Figure 16, upper right quadrant, UR), or necrosis, (PI positive only Figure 16, upper left quadrant, UL). A representative experiment with pugnin A and B is given in Figure 16 below [113].

The cytotoxicity test, MTT, showed a diminution of the viability on HaCaT and PC3 cells treated with pugnin A and B at 100 µM; the variations of mitochondrial membrane potential with DIOC_6_ and cytoplasmic membrane damage with PI was determined (Figure 16). After 24 h of treating HaCaT cells with pugnin A 150 µM, it was observed that 64.4% of the cells were positive for PI and DIOC_6_. The cells can only be stained with PI when the cytoplasmic membrane is damaged, so these results suggest that the treatment with both pugnins affected the integrity of the cytoplasmic membrane and retained their mitochondrial potential, because the cells remained DIOC_6_ positive.

Only a small percentage of necrotic cells were observed when HaCaT cells were treated with pugnin A at 150 µM (7.25%) and 14.7% when pugnin B was tested; however, this percentage was higher than PC-3, indicating that HaCaT cells were more susceptible to the possible cytoplasmic membrane effect of the pugnins. PC3 cells treated with pugnin A and B at 150 µM also showed a DIOC_6_ positive signal, meaning that the mitochondrial membrane potential was not affected by this treatment. The possible effect of this pugnins is thus not mediated by the mitochondria, but is related to the integrity of the cytoplasmic membrane.

PC3 tumor cells treated with both pugnins only showed cells DIOC_6_+/PI− (live cells) and DIOC_6_+/PI+ (apoptotic cells) (Figure 16). This result suggests that the mitochondria were not affected by these peptides, because none of these cells lost the DIOC_6_ stain, thus demonstrating that the mitochondrial potential was not affected. Alternatively, HaCaT cells, showed some cells with DIOC_6_−/PI+ (necrotic). Those results suggest a difference from PC3, and that the mitochondria of HaCaT cells could be affected by the treatment with the pugnins, as these cells lose the stain with DIOC_6_. Taken together, the pugnins showed higher effects against HaCaT cells compared with PC3.

### 3.9. Effect on Apoptosis Induction Evaluated by AnnexinV-PE/SYTOX in the HaCaT and PC3 Cell Line

Flow cytometry was used to assess cell death induced by the pugnin A and B peptide treatments on HaCaT and PC3 cells, using Annexin V-PE/SYTOX staining for 24 h of post-treatment to determine the percentages of apoptotic and necrotic cells (Figure 17). Cell lines negative for both SYTOX and Annexin V staining were live cells (in the lower left quadrant; Q4), while SYTOX-negative Annexin V-positive cells were early apoptotic cells (in the lower right quadrant; Q3). SYTOX-positive Annexin V-positive cells were primarily late apoptotic/necrotic cells (in the upper right quadrant; Q2), and the SYTOX-positive but Annexin V-negative cells were necrotic cells (in the upper left quadrant; Q1) [114].

The treatment with both peptides increased the percentage of cells on apoptosis and necrosisL HaCaT cells treated with pugnin A at 150 µM present 15.2% of apoptotic cells compared with 0.790% that presented no treated cells. Pugnin B at 150 µM concentration showed 17.8% of apoptotic cells and 55.8% of necrotic cells, while pugnin A at the same concentration showed 16.5% of necrotic cells. Thus, pugnin B can cause more cell death by necrosis than pugnin A at 150 µM concentration on HaCaT cells. Detection of greater amounts of SYTOX stained dead HaCaT cells were also observed in the pugnin B 150 µM treatment group, thus suggesting that the cell death mechanisms triggered by this peptide may be different compared to pugnin A, at the same concentration.

When PC3 tumor cells were treated with pugnin A at 150 µM, they showed 17% of apoptotic cells and 13.4% of necrotic cells, and treatment with pugnin B at the same concentration showed 0.062% of apoptotic cells and 28% of necrotic cells. Control cells without treatment showed 0.034% of apoptotic cells and 0.366% of necrotic cells. These results showed that treatment with both peptides increased the percentage of cell death by apoptosis and necrosis on PC3 tumor cells too. HaCaT cells treated with pugnin B at 150 µM concentration showed a higher percentage of necrotic cells than PC3 tumor cells, indicating that this peptide had a stronger effect on the control cells compared with the tumor cell line.

The HaCaT cell line was also used to test the selectivity of the pugnin A and B peptides against tumor cells versus normal cells. Treatment of HaCaT cell lines with both peptides caused a raise in the percentage of cells on apoptosis and necrosis (Figure 17). The percentage of viable cells observed decreased statistically when HaCaT cells were treated with pugnin A at 150 µM (*p* = 0.0018), and when the cells were treated with pugnin B at 50 µM (*p* = 0.0162), 100 µM (*p* = 0.0027), and 150 µM (*p* = 0.0004) concentrations, respectively. The effects of treatment with pugnin B on HaCaT cells also varied from the lower to the maximum concentration of this peptide. On the tumor cell line, PC3 showed more cells with necrosis than with apoptosis after 24 h of treatment with both peptides (*p* < 0.0001). The percentage of viable cells also diminished after the treatment with both peptides (*p* < 0.0001). These results suggest that pugnin A and B showed a non-specific effect on tumoral versus normal cells.

### 3.10. Analysis of the Cell Cycle of HaCaT and PC3 Cell Lines Treated with Pugnin A and B

The cell cycle phases by the PI staining and DNA content analysis were investigated. Results showed that PI could enter the cells after permeabilization with ethanol to bind DNA. The phases of the cell cycle were defined based on the DNA content, being G1 as 2n and G2/M as 4n. When HaCaT and PC3 cells were analyzed, there was a lot of fragmented chromatin, which made it difficult to establish a cell cycle model (data not shown). The fragmented DNA were excluded, after which the distribution of cells based on their DNA content were analyzed (Figure 18).

The number of cells on G1 was higher on HaCaT cells treated with pugnin B at 100 µM (*p* = 0.0334) and 150 µM (*p* = 0.0410) than on PC3 cells, while the treatment with pugnin A at 150 µM for 24 h caused a diminution on the percentage of cells on the G1 phase (*p* = 0.0019) and also on cells in the S phase (*p* = 0.0263).

After analyses of the distribution of HaCaT and PC3 cells treated with pugnins on the G1, S, and G2/M cell cycle phases, results revealed that the pugnins do not significantly affect cell cycle progression.

## 4. Discussion

Positively charged residues are observed in the pugnins, mainly arginine, followed by lysine, which increases solubility in aqueous solutions, favoring intra- and inter-molecular interactions and contributing to helical stability [115,116]. There is evidence of a relationship between arginine residues and antimicrobial activity, being found in a high proportion in antimicrobial peptides; however, it is not entirely clear what specific properties they provide to antimicrobial peptides [117].

The relationship between the antibacterial/anticancer activity and arginine is given by the side chain of this amino acid, which is capable of forming hydrogen bonds with the surrounding molecules, differentiating with other cationic amino acids such as lysine, which when bonded with aromatic compounds, limits the number of hydrogen bridges available, but with arginine this limitation is not present [117]. The role of arginine in the amino or terminal carboxyl region is to stabilize the helical structure [118]. This preponderance of amino acids such as arginine at the ends of the alpha helix is because they contribute with their side chains to binding with the free NH groups of the helix through saline bridges and also favor the electrostatic interactions with the anionic groups of the phospholipids of the bacterial membrane and cancer cells [119,120].

The literature suggests a range of hydrophobicity between 40% and 60% to facilitate the electrostatic interaction of the peptide with the membrane [49,121], The hydrophobicity of the pugnins is an average of 43%, which indicates a good probability of interacting with the membranes to facilitate the insertion of the peptide into them.

Helicity, hydrophobicity, and amphiphilicity are important for the antibacterial and anticarcinogenic activity of peptides [24]. With high hydrophobicity (>76%), the antimicrobial effect is lower due to a strong interaction with phospholipids and also low selectivity to the type of membrane, causing cytotoxic effects in normal cells [122]. It was also found that high hydrophobicities make it difficult for peptides to dissolve; however, although hydrophobicities of pugnins are in the range of 40 to 60%, only pugnin B presented dissolution problems [123].

This is because it can form hydrophobic interactions in the IFWVI residues, according to aggregation probability analysis with the TANGO online tool. These interactions can affect activity as interpeptide aggregation increases the energy cost for the peptide to embed itself in the bacterial cell membrane [104] and can lower the selectivity to the type of membrane [24] case occurred by affecting both HaCaT and PC3 cells.

In the results of the antibacterial test, a trend of varied pugnin selectivity was found, i.e., pugnin A affects *E. faecalis*, *P. aeruginosa*, while pugnin B affects *E. coli* and *S. aureus*. The effect towards Gram-negative bacteria could be due to the variation of the lipid composition of the *E. coli* membrane, having anionic (80%) and zwitterionic (12%) lipids. This probably favored the interaction of pugnins and this membrane, inducing the lateral separation of phospholipids [63]. This lateral separation consists of an interaction of the polar heads of the phospholipids with the cationic residues of the peptide, which facilitates the insertion of the peptide into the membrane by laterally displacing the phospholipids and rearranging the anionic lipids in a separate domain [124]. This reorganization of the lipids around the domain leaves defects in the membrane (pores) and also occurs in recruitment of anionic lipids from other locations on the membrane, where they are necessary for the proper functioning of the membrane, ending up by disturbing the existing domains on the membrane, which negatively affects the bacteria [124].

The minor effect of pugnin A on *S. aureus* could be due to differences in membrane composition concerning Gram-negative ones, as it varies widely among bacteria affecting peptide interaction [24,120]. For *E. faecalis*, the antibacterial effect of pugnin B was lower when compared to pugnin A, and this could be explained by two mechanisms of resistance of *E. faecalis*, the first one by the reduced number of phospholipids with negative charge on the surface of the membrane and the decrease of the fluidity of the membrane, that is, an increase of the rigidity that makes the membrane impenetrable [125]. Another mechanism of resistance by *E. faecalis* is represented by proteases that contribute to the elimination of cationic helical peptides [126].

On the other hand, pugnin B presented a reduced selectivity in terms of cytotoxic effect, because this peptide does not discriminate between cancerous and normal cells which makes it toxic, a condition that occurs very frequently with helical cationic peptides, resulting in a problem for clinical application [120]. The selectivity on peptides allows differentiation of bacterial cells, normal eukaryotes, and cancer cells. This difference lies in the composition of the cell membrane; in the case of bacteria, there are negatively charged lipids such as phosphatidyl glycerol and cardiolipin, which similarly occurs with cancer cells with the presence of anionic lipids such as phosphatidyl serine [120]. In contrast, normal cells have cholesterol, with a predominance of zwitterionic lipids such as phosphatidylcholine [120].

One way to improve the selectivity of peptides is through the net charge. Net charges between +5 and +9 are recognized as the preferred range for cationic peptides. Values above +9 decrease selectivity and can damage human cells [48,49,50], However, in the hemolytic and cellular cytotoxicity analyses of this investigation, damage by pugnins A and B is observed, despite presenting net charge within the mentioned values, which shows that the net charge is not sufficient to improve the selective association to the membrane [127]. This low selectivity may be due, in part, to the high affinity of pugnins A and B for anionic lipids rather than zwitterionics in the eukaryotic cell membrane. Therefore, in order for a peptide to be considered as a therapeutic option, it must not only possess antibacterial properties but also low hemolytic and cytotoxic activities [128].

In the case of the erythrocyte membrane, its phospholipid composition is dominated by phosphatidylcholine, followed by phosphatidylethanolamine and phosphatidylserine, which represents 20% of the eukaryotic cell membrane [129,130].

This cytotoxicity can be attributed to tryptophan in pugnin B which has a strong affinity to the choline heads of the bilayer of HaCaT cells [117]. The hydrophobicity of peptides together with the increased fluidity of the membrane in cancer cells are the contributing factors to their absorption into the membrane [131]. Membrane fluidity is mediated by cholesterol and becomes a selectivity factor between normal and tumor cell membranes, since cholesterol restricts fluidity, and in tumor cells there is less cholesterol; therefore, there is greater fluidity in the membrane facilitating the permeation of anticancer peptides [24]. However, certain tumors have been found to have high cholesterol content in their membranes, for example, breast and prostate cancer cells that become difficult therapeutic targets for peptide use [24].

The diminished fluidity in PC3 did not prevent their damage, mainly by pugnin B. This could be due to the presence of IFWVI residues that indicate the mechanism that favors membrane disruption in both bacterial cells, normal eukaryotic cells, and cancerous eukaryotic cells, according to the results found. The aggregate formation is a property of amphipathic peptides that has received little attention about their activity and selectivity [127]. In this case, pugnin B has a high probability of generating aggregates that manifest themselves with increased cytotoxic activity in tumor cells; however, aggregation may also contribute to the loss of selectivity by causing necrosis in normal eukaryotic cells. In research on [122,132], the same effect on selectivity that occurred in this work is noted.

The effect of anticancer peptides studied to date show a lytic effect on the cell membrane (necrosis) and on the mitochondrial membrane (apoptosis), together with non-membranoid activities [21]. In this study, the mechanism visualized in the test with Anexin V and DIOC_6_ elucidated that the effect of the pugnins is mainly of damage to the cellular membrane, that of inducer of apoptosis and of mitochondrial damage, as much against the normal cells HaCaT as against the PC3.

## 5. Conclusions

This work emerges as an indicator of the potential use of the transcriptome of *B. pugnax* to find peptides with antibacterial–anticancer activity and the potential of transmembrane proteins as a source of sequences for the development of new peptides with alpha-helix cationic characteristics, which are stable bacteria membranes and have highly flexible residues. Here, two pugnins were carefully selected from the *B. pugnax* transcriptome using machine learning algorithms. Subsequently, by MD It was revealed that positively charged residues such as arginine and lysine played an important role in the interaction with water molecules and anionic phospholipids in model membranes of *S. aureus* and *E. coli.* Pugnins A and B exhibit random coil structure in water and upon contact with membranes tend to form the helical structure. Loss of helicity was also found to correlate with changes in peptide activity on the aforementioned membranes.

In order to complement the results obtained by MD simulation, the antibacterial activity against Gram + and Gram - bacteria was evaluated, and a higher selectivity of both pugnins towards Gram - bacteria was observed, highlighting the activity of pugnin A. Similarly, this peptide exhibited the highest antimicrobial and cytotoxic activity in erythrocytes, keratinocytes, and tumor cell lines. Although the antimicrobial and anticancer activity of this peptide has a variable level of efficacy, the peptide shows a selective antitumor activity towards human prostatic cancer cells (PC3), which would be attractive as a template for further structure-activity improvement and studies.

## Figures and Tables

**Figure 1 pharmaceutics-13-00578-f001:**
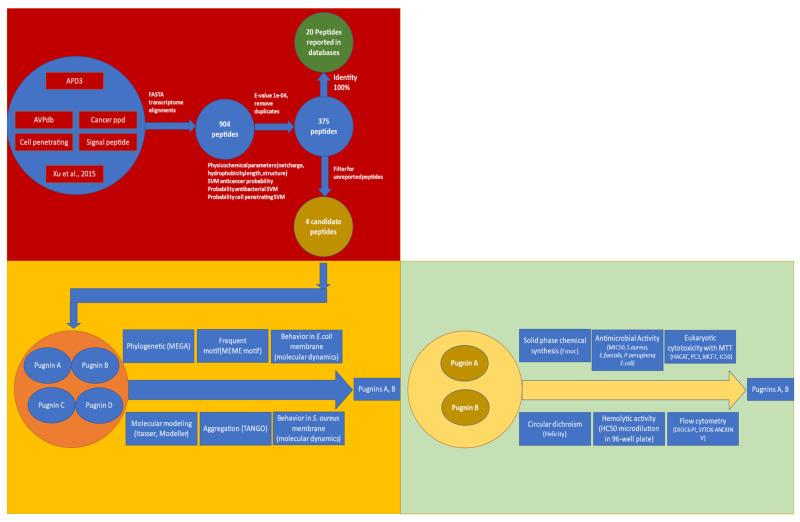
Flow diagram for obtaining pugnins A and B with dual antibacterial and anticancer activity.

**Figure 2 pharmaceutics-13-00578-f002:**
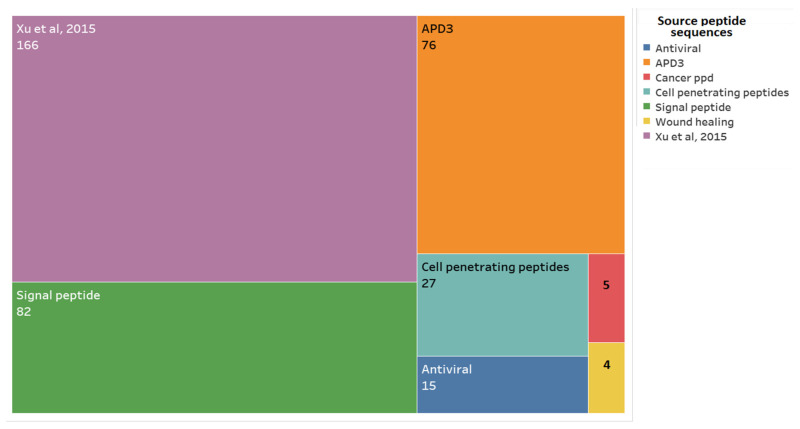
Databases used to query for the 375 putative peptides in the *B. pugnax* transcriptome. The number of peptides found in relation to each database is as shown.

**Figure 3 pharmaceutics-13-00578-f003:**
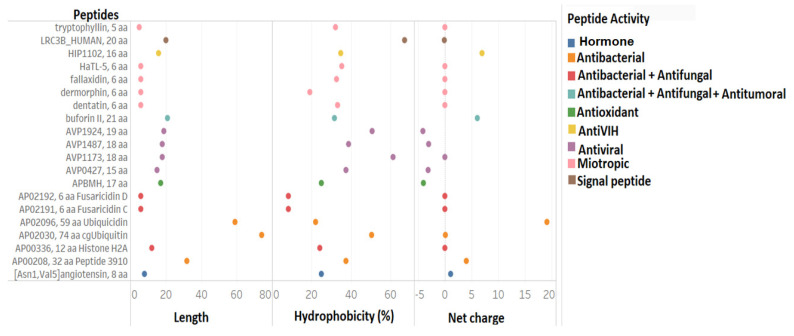
The 20 peptides with 100% identity and similarity, found in the transcriptome of the *B. pugnax* skin. The peptides are related to the physicochemical parameters: length, hydrophobicity, and net charge, as well as to the activity reported by the database.

**Figure 4 pharmaceutics-13-00578-f004:**
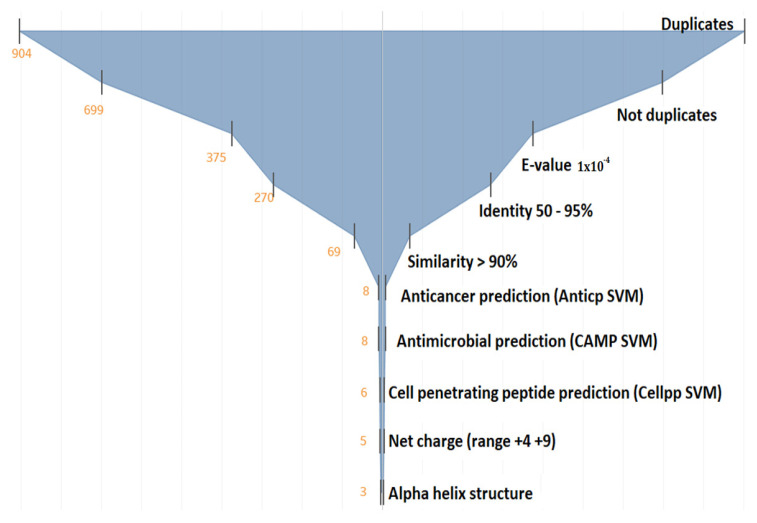
Funnel diagram of the criteria used to select peptide candidates with probable combined antibacterial–anticancer activity (ABC). Anticp: Anticancer peptide. CAMP (Collection of Anti-microbial Peptides), Cellpp (Cell penetrating peptides). SVM (support vector machine algorithm).

**Figure 5 pharmaceutics-13-00578-f005:**
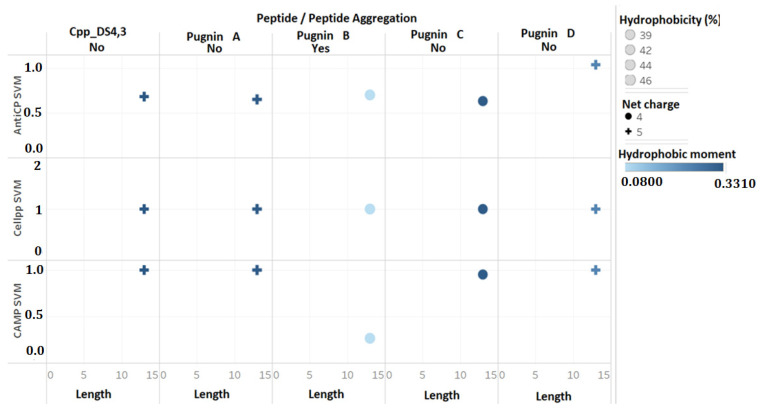
Comparison of physicochemical parameters (net charge, aggregation, hydrophobicity, hydrophobic moment, and length) and SVM prediction between pugnins and peptide Cpp_DS4.3. AntiCP = anticancer SVM prediction, Cellpp = cell-penetrating activity SVM prediction, and CAMP = antimicrobial SVM prediction. +: net charge +5. Blue point: net charge +4.

**Figure 6 pharmaceutics-13-00578-f006:**
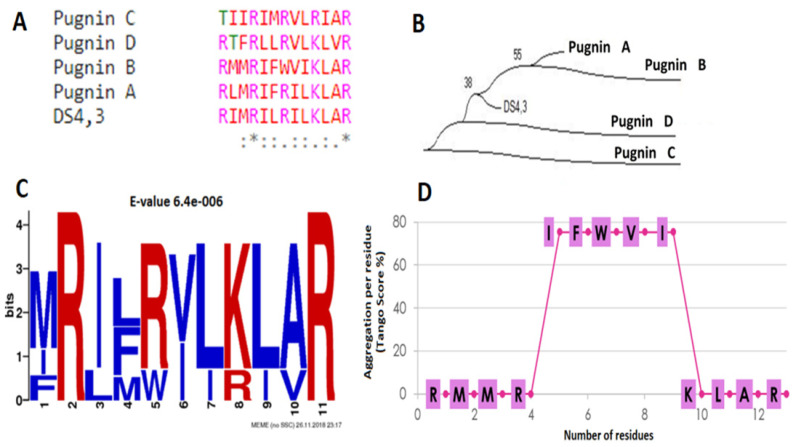
(**A**) Alignment between the pugnins of *B. pugnax* and the DS4.3 peptide. (**B**) Phylogenetic tree of the pugnins of *B. pugnax* and the DS4.3 peptide using neighbor-joining algorithm with the Bootstrap method with 1000 replicates. (**C**) Motif found between the pugnins of *B. pugnax* and the DS4.3 peptide. (**D**) Aggregation profile of pugnin B using the TANGO algorithm [46] with the following parameters: 37 °C, pH 7.2, and ionic strength 0.02 M.

**Figure 7 pharmaceutics-13-00578-f007:**
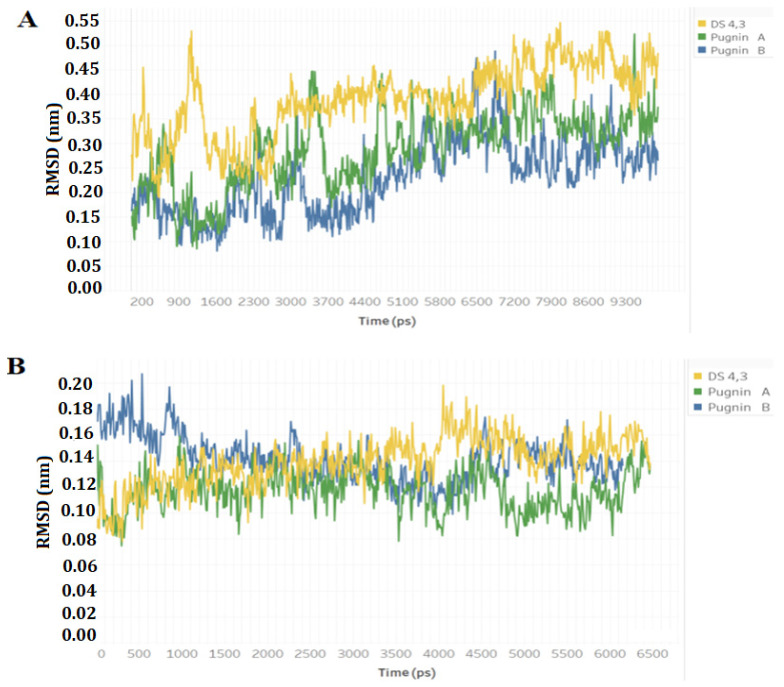
Root mean squared deviation (RMSD) of the peptides in water and the *E. coli* membrane model. (**A**) Comparison of the RMSD of the original peptide DS4.3, pugnin A and B in water. (**B**) Comparison of the RMSD of the original peptide DS4.3, pugnin A and B in the *E. coli* membrane model.

**Figure 8 pharmaceutics-13-00578-f008:**
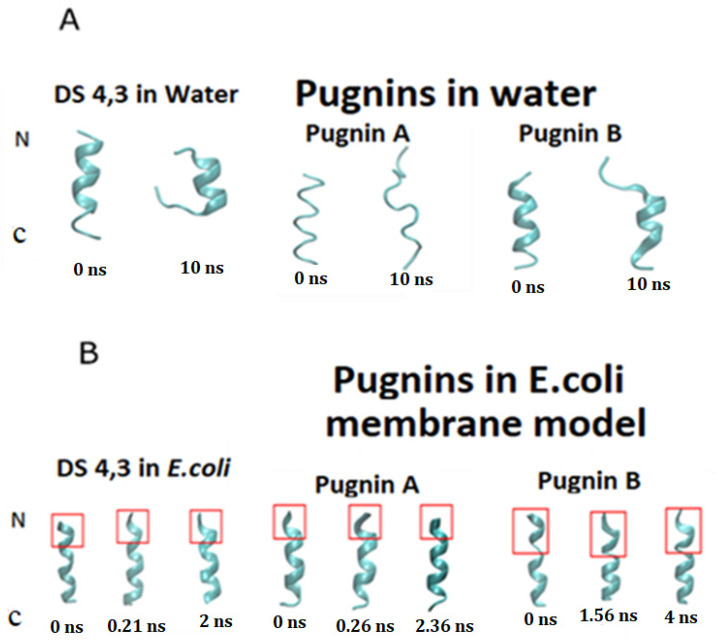
3D structures obtained from simulation with molecular dynamics at different times in nanoseconds (ns). (**A**) 3D structures of peptides in water. (**B**) 3D structures of the peptides in the *E. coli* membrane model. The most unstable regions in the helical structure are detailed in red.

**Figure 9 pharmaceutics-13-00578-f009:**
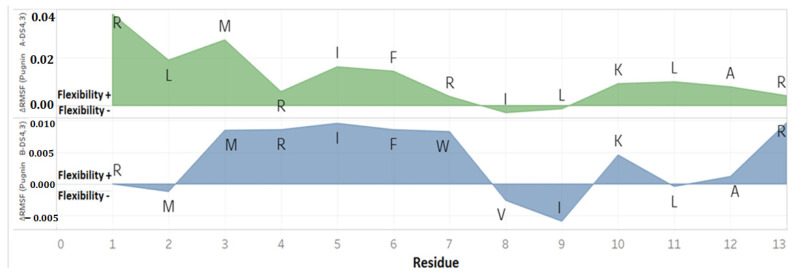
∆RMSF (nm) between the DS4.3 peptide and the pugnins, where the residues of each pugnin can be seen in the positive region as flexibility+ or in the negative region as flexibility−.

**Figure 10 pharmaceutics-13-00578-f010:**
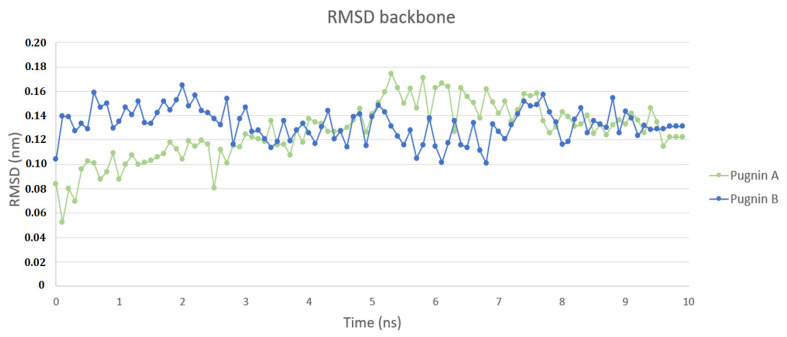
Root mean squared deviation (RMSD) of the pugnins in *S.aureus* membrane model.

**Figure 11 pharmaceutics-13-00578-f011:**
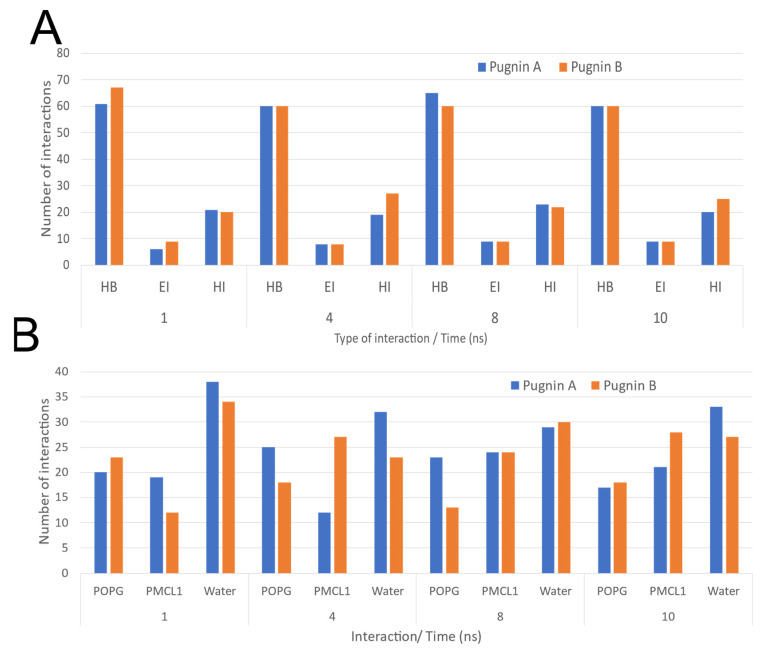
Interactions between pugnins with the *S.aureus* Membrane Model System during the 10 ns. (**A**) Interactions of pugnins in total at 1, 4, 8 and 10 ns. (**B**) Interaction of pugnins with phospholipids and water at 1, 4, 8 and 10 ns. HB: hydrogen bond; EI: electrostatic interaction; HI: hydrophobic interaction; POPG: phosphatidylglycerol; PMLC1: cardiolipin.

**Figure 12 pharmaceutics-13-00578-f012:**
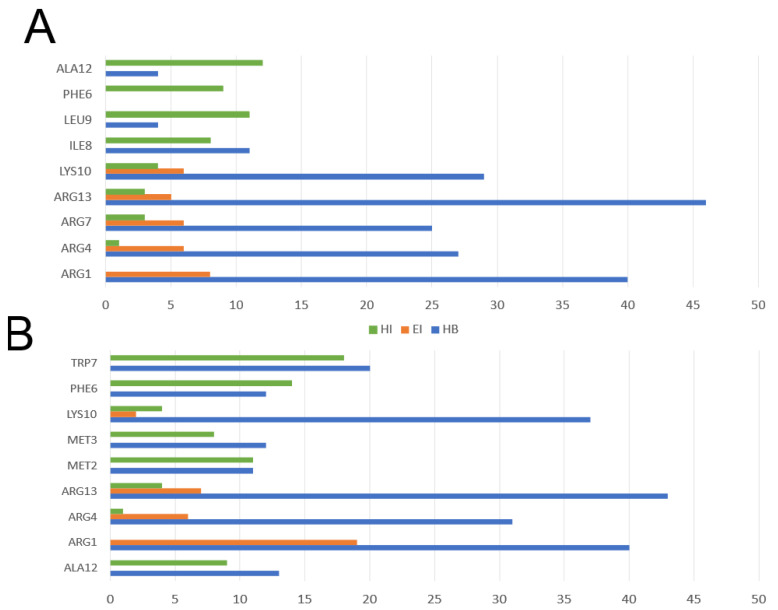
Residues with the highest number of pugnins interactions with the *S. aureus* model membrane system. (**A**) Pugnin A residues that interacted the most with the *S. aureus* model membrane (**B**) Pugnin B residues that interacted the most with the *S. aureus* model membrane. HB: hydrogen bond; EI: electrostatic interaction; HI: hydrophobic interaction.

**Figure 13 pharmaceutics-13-00578-f013:**
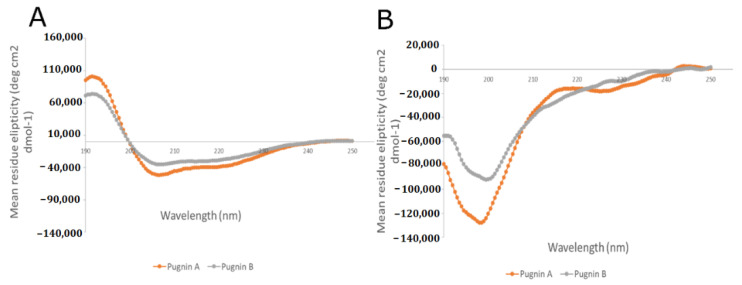
Circular dichroism of the pugnins in TFE 30% (**A**) and pugnins in water (**B**). (See Appendix A).

**Figure 14 pharmaceutics-13-00578-f014:**
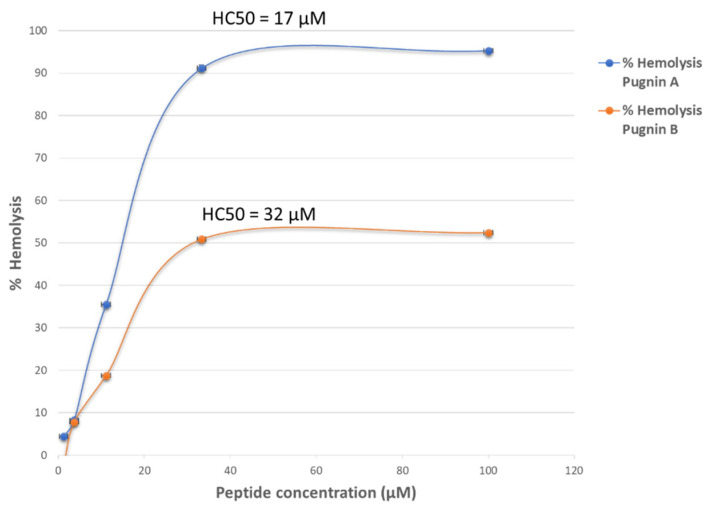
Percentage of hemolytic activity of the three pugnins A and B, with their respective hemolytic concentrations 50 (HC50).

**Figure 15 pharmaceutics-13-00578-f015:**
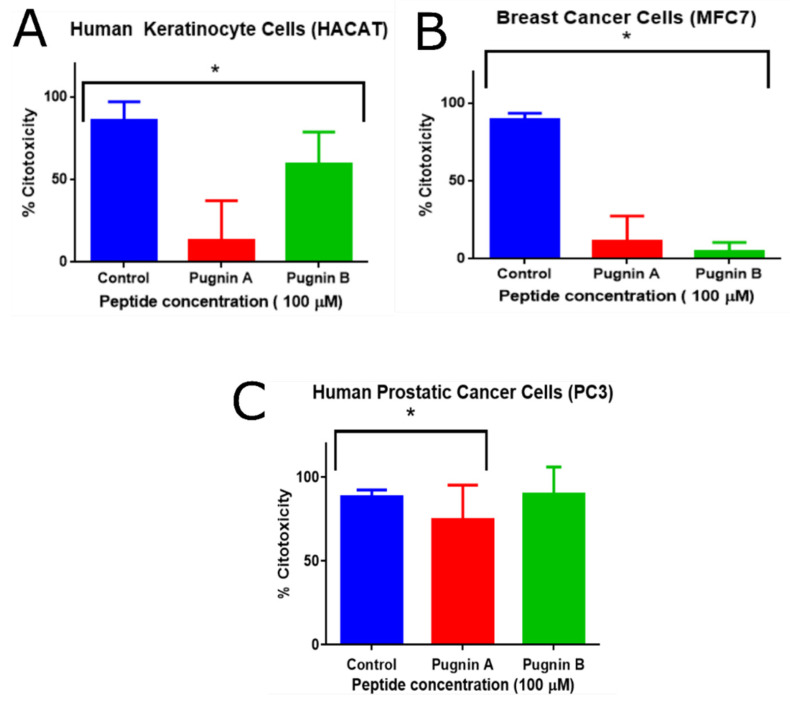
Cytotoxicity of pugnin A and B at 100 µM versus the positive control (Triton X-100). (**A**) Cytotoxicity of peptides against human keratinocytes. (**B**) Cytotoxicity of peptides against breast cancer cells (MFC7). (**C**) Cytotoxicity of peptides against prostate cancer cells (PC3). The asterisk symbolizes a statistically significant difference between the control and the treatments, with *p* < 0.05.

**Figure 16 pharmaceutics-13-00578-f016:**
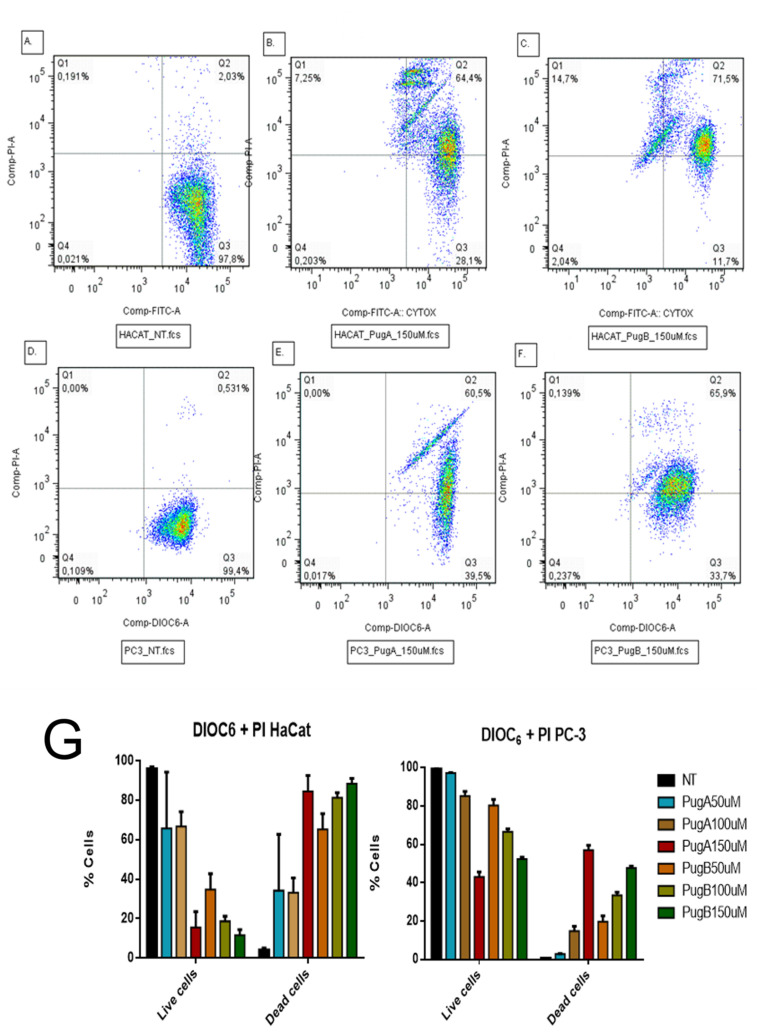
Variations of mitochondrial membrane potential with DIOC^6^ and cytoplasmic membrane damage with PI analyzed by flow cytometry. HaCaT and PC3 cells were treated with pugnin A and B at concentrations of 50, 100, and 150 µM for 24 h. On the figure we only showed 150 µM. (**A**) HaCaT without treatment (control cells). (**B**) HaCaT treated with pugnin A 150 µM for 24 h. (**C**) HaCaT treated with pugnin B 150 µM for 24 h. (**D**) PC3 without treatment (control cells). (**E**) PC3 treated with pugnin A 150 µM for 24 h. (**F**) PC3 treated with pugnin B 150 µM for 24 h. (**G**) Histograms show the percentage of live and dead cells after 24 h of treatment with pugnin A and B. Those histograms are the quantification from the flow cytometry. Three independent experiments were conducted for each cell line. The data are presented as mean ± SEM.

**Figure 17 pharmaceutics-13-00578-f017:**
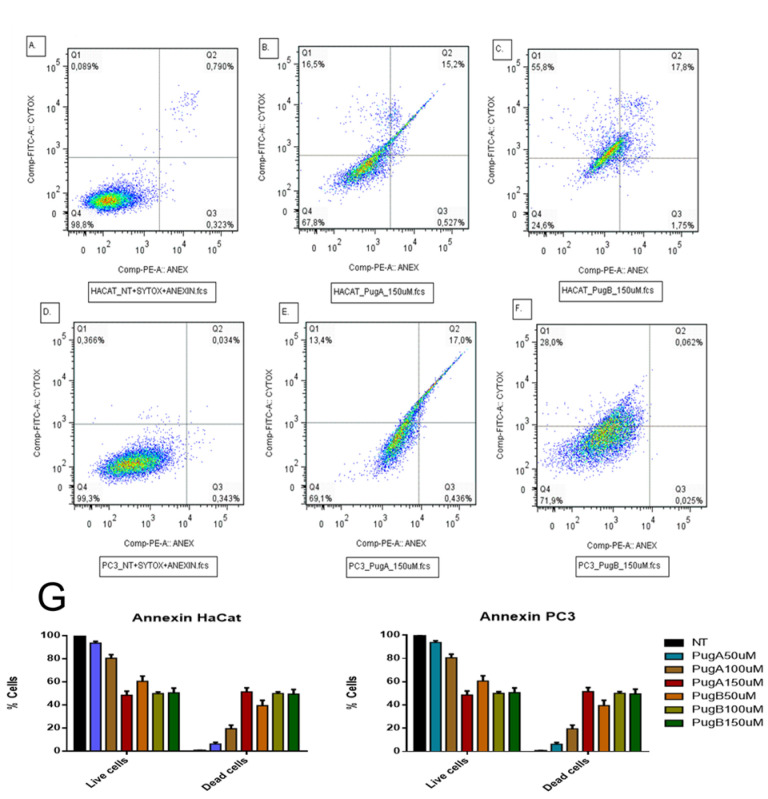
Apoptosis and necrosis analysis by flow cytometry using Annexin V-PE/SYTOX double staining. HaCaT and PC3 cells were treated with pugnin A and B at concentrations of 50, 100, and 150 µM for 24 h. On the figure we only showed 150 µM. (**A**) HaCaT without treatment (control cells). (**B**) HaCaT treated with pugnin A 150 µM for 24 h. (**C**) HaCaT treated with pugnin B 150 µM for 24 h. (**D**) PC3 without treatment (control cells). (**E**) PC3 treated with pugnin A 150 µM for 24 h. (**F**) PC3 treated with pugnin B 150 µM for 24 h. Three independent experiments were carried out for each cell line. (**G**) Histograms show the percentage of live and dead cells after 24 h of treatment with pugnin A and B. Those histograms are the quantification from the flow cytometry. Three independent experiments were carried out for each cell line. The data are presented as mean ± SEM.

**Figure 18 pharmaceutics-13-00578-f018:**
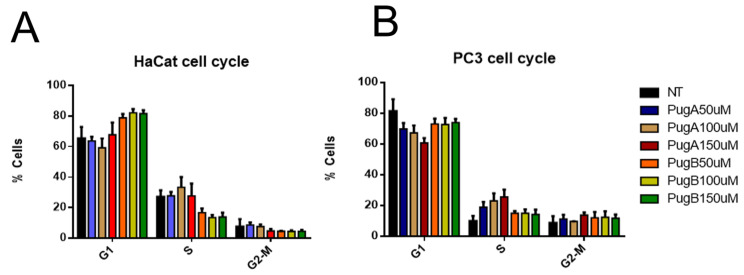
The percentage of cells on phase G1, S, and G2/M based on the DNA content measured using PI staining. (**A**) HaCat cells (**B**). PC3 cells. The cells were treated with the respective pugnin at 50, 100, and 150 µM for 24 h and then fixed in 70% ethanol, resuspended in PBS, and stained with PI.

**Table 1 pharmaceutics-13-00578-t001:** Antibacterial test with pugnins.

Inhibitory Concentration (µM)												
Peptides	*S. aureus ATCC 25923*			*E. faecalis ATCC 29212*			*P. aeruginosa ATCC 27853*			*E. coli ATCC 25922*		
	MIC50	MIC90	MBC	MIC50	MIC90	MBC	MIC50	MIC90	MBC	MIC50	MIC90	MBC
Pugnin A	11.0	148.9	183.4	0.70	107.2	150.0	4.1	28.5	34.6	14.2	69.4	83.3
Pugnin B	9.20	549.7	684.8	18.0	111.2	134.5	8.9	158.4	195.7	0.10	57.8	72.2

## Data Availability

Repository name: Sequence Read Archive Data identification number: SRP151854 Direct URL to data: https://www.ncbi.nlm.nih.gov/sra/?term=boana±pugnax Repository name: Transcriptome Shotgun Assembly Data identification number: GINY010000001 Direct URL to data: https://www.ncbi.nlm.nih.gov/Traces/wgs/GHME01?val=GINY01_accs.

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
