# Peer review of "In Silico Selection and Evaluation of Pugnins with Antibacterial and Anticancer Activity Using Skin Transcriptome of Treefrog (*Boana pugnax*)"

_pharmaceutics, 2021, doi:10.3390/pharmaceutics13040578_

Round 1
Reviewer 1 Report
The authors report the selection and evaluation of pugnins with antibacterial and anticancer activity. The paper is well-conceived and written.
However, the correlation between anticancer and antimicrobial activities of peptides should be better explained in the introduction.
In the experimental design, the authors choose only the gram-negative bacterial membrane model for molecular dynamics studies. The authors should explain the reason since pugnins showed interesting effects also on Gram positive bacteria.
Author Response
Response to Reviewers Comments
April 4th, 2021
Executive Editor
Molecules
Dear Executive Editor,
To facilitate the review of our responses, the following is a point-by-point response to the comments of reviewers. The changes are highlighted in yellow in the text.
REVIEWER 1 COMMENTS
We appreciate the reviewer’s comments.
Reviewer: The authors report the selection and evaluation of pugnins with antibacterial and anticancer activity. The paper is well-conceived and written.
However, the correlation between anticancer and antimicrobial activities of peptides should be better explained in the introduction.
Response: We made the addition of a paragraph in the introduction, where we explained that when using cationic peptides against bacteria, due to the anionic charge of phospholipids in both bacteria and cancer cells this class of peptides could have a cytotoxic effect. Lines 79-86.
Reviewer: In the experimental design, the authors choose only the gram-negative bacterial membrane model for molecular dynamics studies. The authors should explain the reason since pugnins showed interesting effects also on Gram positive bacteria.
Response: In the first instance, we chose the E. coli membrane as a model system because there were lower MICs than S. aureus. But in order to differentiate between the cytotoxic effect of pugnin A and B we decided to perform molecular dynamics and understand what type of interaction was occurring. Lines 573-608.
Reviewer 2 Report
The work studied new antimicrobial and anticancer peptides from the Boana pugnax transcriptoma by searching in several peptide databases and publications. The study is very extensive and detailed, bringing detailed in silico and in vitro studies. There are several concerns about the manuscript experiments, which it is pointed out below. Also, there are some questions about experiments and the work state-of-art that must be improved, as follows:
- The discussion is very long and do not support entirely the paper hypothesis.
- In terms of therapeutic index, it is worthy to carry on further studies of these two chosen peptides? They are very toxic and the MICs are, in mostly cases, high; also, the hemolytic activity is lower than several MICs presented.
- It is necessary to demonstrate if the pugnins A and B have helicity and quantifies it by CD technique, in order to improve the discussion (see notes below).
- The theme is very interesting, but the authors should better explain the methodology pathways and decisions.
- It is mandatory a extensive english proofreading for readability improvement.
Points to correct:
Abstract
Line 23: "...stability in the aqueous medium..."
Line 24: "...using molecular dynamics. These pugnins..." and remove "the"
Line 27: "... to evaluate their..."
Manuscript
Line 54: "Boana pugnax", since it appears for the first time in the text
Ref. 13: Page Not Found
Line 121: "from the database"
Line 130: site is invalid. Please verify.
Line 139: "physicochemical"
Line 141-145: The phrase is too long, difficulting the interpretation. Divide into small and objetives statements.
Line 141: What it CAMP? cationic antimicrobial peptides? If yes, it must be described the abbreviation meaning at the first time it appears in the text, as well the others abbreviations mentioned, such as Cellpp and Anticp.
Line 160: "were attached", avoid word repetition.
Line 162: "helix structure, since peptides"
Line 163: remove "that is"
Line 188: RAMPAGE site "The requested URL was not found on this server." Please verify.
Line 225: remove the first abbreviation "VMD ("
Line 253: Correct the number 10^8 of CFU/ml
Line 253-254: Check the sentence: "The minimum inhibitory concentration (MIC50) was calculated using linear..."
Line 261, 266 and 292: Why they diluted the peptide concentrations by a factor of 3 instead of normal dilution, by half?
Line 273: "was calculated, which is the necessary..."
Line 282 and 294: "37°C"
Line 284: "in DMEM medium. The medium was..."
Line 315: Why use different peptide concentrations here?
Line 324: check the number 10^5
Line 325: The authors need to homogenize the liter units in the entire manuscript (the recommended is L)
Line 446: The authors should explain clearly why they decide to synthesize the forth peptide derived from DS4.3 peptide. Also, it was not indicated which were the three peptides chosen.
Figure 4: This figure is confusing. The authors should use the figure 5A, which contains the primary sequence of the peptides to support the parameters discussed in this figure. It will became much more clear and easy to understand.
Line 482: "The presence of C and R motif" - It is not clear why it has this name.
Line 493: Why not studied the 4 peptides? I do not understand why they choose pugnin B for the studies. As explained in section 3.2, pugnins C and D showed better parameters regarding the antimicrobial and anticancer activities, even they are associated with the calcium channel proteins. Also, pugnin B presented the lowest hydrophobic moment, including its potential capacity of aggregation.
Line 500: Check the sentence: "This was then confirmed with RAMPAGE"
Line 506: "such as root mean squared deviation (RMSD)" the abbreviation meaning should appears at the first time in the text (line 230).
Line 510: "DS4,3" or "DS4.3"? It is necessary to homogeneize the peptide name.
Lines 514-517: The statement is too confusing. Please rewrite.
Line 519: E. coli should be in italic.
Figure 7: What does it means the letter "T"? I think it should be C, representing the C-terminal region of the peptides.
Line 552: It will be interesting to measure the GRAVY index for hydrophobicity and present it in Figure 4 or it can be described here in ths section.
Line 558 and 561: the authors must present the chromatograms profiles and mass spectra for each peptide studied in the Supplementary Materials.
Line 568: "they, formed" remove comma.
Line 573: It will be interesting to make CD measurements with micelles (LPC or DMPG or both) at least to analyse their secondary structures in this environment, in order to correlate to the other results.
Line 600: Check the sentence: "with the hemolytic concentration 50 (HC50)" remove "50".
Line 633: Please use the correct molecular formula of DIOC6, with the number subscribed.
Line 745: this sentence is controversial to that affirmed in Line 569. Do the pugnins adopt alpha helices when in contact with membranes? There is no information to support this statement in the paper.
Lines 749 to 753: the sentence is too long. Split it in two or three to facilitate the readability.
Line 768: "...only pugnin B presented dissolution problems"
Author Response
Response to Reviewers Comments
April 4th, 2021
Executive Editor
Molecules
Dear Executive Editor,
To facilitate the review of our responses, the following is a point-by-point response to the comments of reviewers. The changes are highlighted in yellow in the text.
REVIEWER 2 COMMENTS
We appreciate the reviewer’s comments.
The work studied new antimicrobial and anticancer peptides from the Boana pugnax transcriptoma by searching in several peptide databases and publications. The study is very extensive and detailed, bringing detailed in silico and in vitro studies. There are several concerns about the manuscript experiments, which it is pointed out below. Also, there are some questions about experiments and the work state-of-art that must be improved, as follows:
- The discussion is very long and do not support entirely the paper hypothesis.
- In terms of therapeutic index, it is worthy to carry on further studies of these two chosen peptides? They are very toxic and the MICs are, in mostly cases, high; also, the hemolytic activity is lower than several MICs presented.
- It is necessary to demonstrate if the pugnins A and B have helicity and quantifies it by CD technique, in order to improve the discussion (see notes below).
- The theme is very interesting, but the authors should better explain the methodology pathways and decisions.
- It is mandatory a extensive english proofreading for readability improvement.
Points to correct:
Abstract
Line 23: "...stability in the aqueous medium..."
Response: It was corrected to in. Line 23.
Line 24: "...using molecular dynamics. These pugnins..." and remove "the"
Response: It was corrected to “. These” and “the via” was changed to “via”. Line 24.
Line 27: "... to evaluate their..."
Response: It was corrected “show” to “evaluate”. Line 27.
Manuscript
Line 54: "Boana pugnax", since it appears for the first time in the text
Response: It was corrected “B.pugnax” to “Boana pugnax”. Line 54.
Ref. 13: Page Not Found
Response: The DOI was corrected to http://dx.doi.org/10.20517/cdr.2018.03
Line 121: "from the database"
Response: It was corrected from “to” to “from the”. Line 121.
Line 130: site is invalid. Please verify.
Response: New site https://npsa-prabi.ibcp.fr/cgi-bin/npsa_automat.pl?page=/NPSA/npsa_server.html . Line 136.
Line 139: "physicochemical"
Response: It was corrected from “Physicochemical” to “physicochemical”. Line 145.
Line 141-145: The phrase is too long, difficulting the interpretation. Divide into small and objetives statements.
Response: It was corrected from 1 statement to two statements. Line 149.
Line 141: What it CAMP? cationic antimicrobial peptides? If yes, it must be described the abbreviation meaning at the first time it appears in the text, as well the others abbreviations mentioned, such as Cellpp and Anticp.
Response: It was corrected to “Collection of Anti-microbial Peptides (CAMP) prediction SVM was used, for cell-penetrating peptide we worked with Cell penetrating peptide (Cellpp) SVM and Anticancer peptide (Anticp)” Lines 147-149.
Line 160: "were attached", avoid word repetition.
Response: It was corrected to “were attached” Lines 167.
Line 162: "helix structure, since peptides"
Response: It was corrected to “helix structure, since peptides” Lines 169.
Line 163: remove "that is"
Response: It was removed “that is” Lines 169.
Line 188: RAMPAGE site "The requested URL was not found on this server." Please verify.
Response: The page does not work anymore, so we decided to change it to Molprobity (http://molprobity.biochem.duke.edu/). We also added the corresponding literature “Chen, V.B.; Arendall, W.B.; Headd, J.J.; Keedy, D.A.; Immormino, R.M.; Kapral, G.J.; Murray, L.W.; Richardson, J.S.; Richardson, D.C. MolProbity : All-Atom Structure Validation for Macromolecular Crystallography. Acta Crystallogr D Biol Crystallogr 2010, 66, 12–21, doi:10.1107/S0907444909042073.”. Lines 195-196.
Line 225: remove the first abbreviation "VMD ("
Response: It was removed changed to “Visual Molecular Dynamics (VMD)”. Line 233.
Line 253: Correct the number 10^8 of CFU/ml
Response: It was removed changed to “1-4 x 108 colony”. Line 263.
Line 253-254: Check the sentence: "The minimum inhibitory concentration (MIC50) was calculated using linear..."
Response: It was corrected from “The minimum inhibitory concentration that was calculated was the MIC50 using lineal regression with the equation of the line” to “The minimum inhibitory concentration 50 (MIC50) was calculated using lineal regression”. Line
Line 261, 266 and 292: Why they diluted the peptide concentrations by a factor of 3 instead of normal dilution, by half?
Response: Because we were interested in an antibacterial activity of the peptide below 30 uM, we decided to make the dilution by a factor of three to be sure that it encompassed a value closer to this.
Line 273: "was calculated, which is the necessary..."
Response: It was corrected to “was calculated, which is the necessary”. Line 282.
Line 282 and 294: "37°C"
Response: It was corrected in Line 291,295, 303.
Line 284: "in DMEM medium. The medium was..."
Response: It was corrected to “in DMEM medium. The medium was supplemented”. Line 293.
Line 315: Why use different peptide concentrations here?
Response: Because we did a pre-test at those concentrations and they gave us values higher than 33.33 uM. Therefore, we increased the range up to 150 uM to observe more clearly a cytotoxic effect of the peptides. The lethality of 150uM was unknown since peptides were evaluated up to 100uM by MTT. In this way the concentration of peptides was raised in the flow cytometry assay in order to see the increased cytotoxicity they could cause in cancer cells.
Line 324: check the number 10^5
Response: It was corrected to “1.5 x 105”. Line 333.
Line 325: The authors need to homogenize the liter units in the entire manuscript (the recommended is L)
Response: It was corrected. Lines 263, 291, 294, 302, 321, 340, 341.
Line 446: The authors should explain clearly why they decide to synthesize the forth peptide derived from DS4.3 peptide. Also, it was not indicated which were the three peptides chosen.
Response: A paragraph was added explain it much better. Line 469 – 474.
Figure 4: This figure is confusing. The authors should use the figure 5A, which contains the primary sequence of the peptides to support the parameters discussed in this figure. It will became much more clear and easy to understand.
Response: We decided to keep this figure because it shows the SVM prediction values that help us to support the choice of the four candidates.
Line 482: "The presence of C and R motif" - It is not clear why it has this name.
Response: It was corrected to “R/KXXR/K motif” to avoid confusión. Lines 509, 511.
Line 493: Why not studied the 4 peptides? I do not understand why they choose pugnin B for the studies. As explained in section 3.2, pugnins C and D showed better parameters regarding the antimicrobial and anticancer activities, even they are associated with the calcium channel proteins. Also, pugnin B presented the lowest hydrophobic moment, including its potential capacity of aggregation.
Response: We decided to test this peptide because it showed high sequence similarity to pugnin A and peptide DS4.3. In addition, aggregation could result in better activity, which we verified by performing MICs. Pugnins C and D gave us MICs above 100uM and no cytotoxic activity against cancer cells.
Line 500: Check the sentence: "This was then confirmed with RAMPAGE"
Response: It was changed to “Molprobity results showed that 100% of the waste was obtained in the favorable region of the alpha-helix formation, indicating that our 3D models have the appropriate parameters for a protein structure”. Lines 527-529.
Line 506: "such as root mean squared deviation (RMSD)" the abbreviation meaning should appears at the first time in the text (line 230).
Response: It was changed. Lines 237-529.
Line 510: "DS4,3" or "DS4.3"? It is necessary to homogeneize the peptide name.
Response: It was corrected. Lines 175, 241, 502, 541, 548.
Lines 514-517: The statement is too confusing. Please rewrite.
Response: It was changed to “Helix stability was lost at the amino and carboxyl ends of pugnins A and B, being conserved only in the region comprising the IFW residues.”. Lines 541-543.
Line 519: E. coli should be in italic.
Response: It was corrected in line 545.
Figure 7: What does it means the letter "T"? I think it should be C, representing the C-terminal region of the peptides.
Response: It was changed to “C” in figure 8.
Line 552: It will be interesting to measure the GRAVY index for hydrophobicity and present it in Figure 4 or it can be described here in this section.
Response: These are GRAVY results for peptides.
|
|
|
GRAVY |
|
Pugnin A |
RLMRIFRILKLAR |
0,385 |
|
Pugnin B |
RMMRIFWVIKLAR |
0,546 |
|
Pugnin C |
TIIRIMRVLRIAR |
0,846 |
|
Pugnin D |
RTFRLLRVLKLVR |
0,292 |
Line 558 and 561: the authors must present the chromatograms profiles and mass spectra for each peptide studied in the Supplementary Materials.
Response: Fueron agregados a material suplementario como figuras 7 y 8. Lines 621, 626.
Line 568: "they, formed" remove comma.
Response: It was changed. Line 632.
Line 573: It will be interesting to make CD measurements with micelles (LPC or DMPG or both) at least to analyse their secondary structures in this environment, in order to correlate to the other results.
Response: In a next job we could do it. Thank you very much.
Line 600: Check the sentence: "with the hemolytic concentration 50 (HC50)" remove "50".
Response: It was removed. Line 665.
Line 633: Please use the correct molecular formula of DIOC6, with the number subscribed.
Response: It was corrected. Line 698.
Line 745: this sentence is controversial to that affirmed in Line 569. Do the pugnins adopt alpha helices when in contact with membranes? There is no information to support this statement in the paper.
Response: “some antimicrobial peptides are random coils in solution, they adopt a helical structure when interacting with the membrane” Line 552-554.
Lines 749 to 753: the sentence is too long. Split it in two or three to facilitate the readability.
Response: We splitted in three. Lines 824-839.
Line 768: "...only pugnin B presented dissolution problems"
Response: It was corrected. Lines 833-834.
Reviewer 3 Report
The authors present original data about antimicrobial peptides discovered by a combination of methodologies, starting from the bioinformatic analysis of transcriptomes
Synthetic peptides were prepared and evaluated in vitro regarding their antimicrobial and anticancer activity. Additionally, the mechanism of cytotoxicity was preliminarily investigated by flow cytometry and related techniques.
An interesting article that deserves publication after extensive improvement.
Concerns and suggestions:
Introduction:
Exclude references 5 and 26 that are the same and contains similar information that in reference 8 that is more recent.
Reference 28 is incomplete
Reference 84 is incomplete
Reference 135 is incomplete
It is recommended to check the other references for their correctness;
Line 68- This affirmation about ABC is too premature since a large path exists in the drug development process;
Line 72- This phrase is quite confusing and peptides with such kind of structural characteristics are not restricted to antimicrobial peptides. It is recommended to reconsider this point and update this and other references as pointed in additional commentaries below.
Line 80 - I think these biological features are not definitively observed in all antimicrobial/anticancer peptides. It is recommended to specify which are able to have these characteristics.
lines 82-83: This paragraph sounds repetitive and it will make sense if the paragraph starting in line 72 is rewrite and text and bibliography redundancies eliminated.
Materials and methods:
Line 101: How was this transcriptome obtained? It is recommended to specify.
Lines 111 to 118: These paragraphs sound confused. It is recommended to detail because is not clear what the authors have done.
Item 2.1.4:
In general, the M&M should be better described and procedures more detailed, even if references were included, to facilitate reading and understand what kind of techniques, or algorithm, etc, was selected for a procedure.
Information on its physicochemical properties, such as net charge, percentage of hydrophobicity, hydrophobic momentum, and sequence length was obtained from the pep-128 tide sequences found in the transcriptome [45–47]. (How?)
item 2.3: It is recommended to include the data of synthetic peptides in a table.
items 2.5 and 2.6: Uniformize the descriptions, sources, and abbreviations of cell lines in the first time they appear in the text.
Item 2.7: In MTT cytotoxic assay, with the selected cell lines, the peptide concentrations were up to 100 microM. But in the flow cytometry assay, the peptide concentrations were up to 150 microM. And in the apoptosis assay and cell cycle analysis, the peptide concentrations were not specified. It is recommended to uniformize such information.
3. Results
In general, the interpretation of "results" should be more concise and less speculative and concentrate the discussion on their own results.
Line 363: This sentence sounds uncomprehensible.
Figure 2, Legend: Is angiotensin a biological activity?
Lines 431-433: This affirmation is contradictory, isn't it?
Figure 3, Legend: For clarity, it is recommended to include a legend with each procedure used to obtain peptides as depicted in this funnel diagram.
For example, describing the meaning of abbreviature procedures between parenthesis;
Line 442: vNa+-channel from what kind of organism?
Line 458: the authors speculated that pugnins might be cell-penetrating. Why this was not experimentally checked along with the other activities? It could be discussed.
Figure 4: Symbols are confusing since "+" appears as positive for a given activity.
Line 481: "alkaline", is this term used for amino acid residues?
Lines 523-524: This sentence sounds strange. What kind of solution?
Lines 526-528: Not necessarily this affirmation is true. Some active AMPs are random coils in solution but adopts helicoidal structure upon membrane interaction.
Line 613: By taking a look at the data, both peptides are only slightly cytotoxic!
Although the pugnins peptides have an apparent anticancer activity, this only is observable with 100 uM of peptide - a concentration several times higher than those concentrations that cause hemolysis. So, these peptide seems, accordingly to experimental data, to be an antimicrobial peptide, not an efficient anticancer peptide.
Line 669: the concentration is relatively high for a promising activity;
Line 672: Again, why the level of cell death caused by peptides in the annexin-V/SYTOX diverged from the MTT assay?
Figure 14 and results: If at 150 uM pugnins cause cell death, what is the rationale to check the phase of the cell cycle of dying cells at lethal doses?
Discussion:
General: Again, it is recommended to the author to be less speculative and make a correlation with their findings.
Based on the present results, pugnins peptides are antimicrobial that work at 10 -20 uM range, but they are slightly cytotoxic, with a residual cytotoxic (anticancer) activity that is effective with such a high concentration of peptides (over 150 uM).
Lines 743-747: The influence of positively charged amino acid residues and the effect on peptides on bacteria have been studied. It is important to update this information.
Line 776: In this part of the discussion, authors could correlate the structure-activity of pugnin A and B and the antibacterial selectivity, regarding cell wall and membrane composition.
Ref 134, Line 813: Does this reference mention pugnins? It is recommended to specify what it refers.
Lines 838-841: For this and the other experimental reasons, pugnins appear as a potential AMP but not a good anticancer peptide.
Conclusion:
This should be re-elaborated and rewritten after the overall amendment of the manuscript to deliver the most relevant findings.
Author Response
Response to Reviewers Comments
April 4th, 2021
Executive Editor
Molecules
Dear Executive Editor,
To facilitate the review of our responses, the following is a point-by-point response to the comments of reviewers. The changes are highlighted in yellow in the text.
REVIEWER 3 COMMENTS
We appreciate the reviewer’s comments.
The authors present original data about antimicrobial peptides discovered by a combination of methodologies, starting from the bioinformatic analysis of transcriptomes
Synthetic peptides were prepared and evaluated in vitro regarding their antimicrobial and anticancer activity. Additionally, the mechanism of cytotoxicity was preliminarily investigated by flow cytometry and related techniques.
An interesting article that deserves publication after extensive improvement.
Concerns and suggestions:
Introduction:
Exclude references 5 and 26 that are the same and contains similar information that in reference 8 that is more recent.
Reference 28 is incomplete
Reference 84 is incomplete
Reference 135 is incomplete
Response: references 5 and 26 were removed. References 28, 84 and 135 were completed.
It is recommended to check the other references for their correctness;
Line 68- This affirmation about ABC is too premature since a large path exists in the drug development process;
Response: We changed word to “could be”. Line 68.
Line 72- This phrase is quite confusing and peptides with such kind of structural characteristics are not restricted to antimicrobial peptides. It is recommended to reconsider this point and update this and other references as pointed in additional commentaries below.
Response: Statement was removed to avoid redundance with the lines 86-89.
Line 80 - I think these biological features are not definitively observed in all antimicrobial/anticancer peptides. It is recommended to specify which are able to have these characteristics.
Response: It was added “R/KXXR/K motif”. Line 82.
lines 82-83: This paragraph sounds repetitive and it will make sense if the paragraph starting in line 72 is rewrite and text and bibliography redundancies eliminated.
Response: Statement was removed to avoid redundance with the lines 86-89.
Materials and methods:
Line 101: How was this transcriptome obtained? It is recommended to specify.
Response: transcriptome retrieval was added. Lines 102-106.
Lines 111 to 118: These paragraphs sound confused. It is recommended to detail because is not clear what the authors have done.
Response: It was corrected. Lines 118 – 120.
Item 2.1.4:
In general, the M&M should be better described and procedures more detailed, even if references were included, to facilitate reading and understand what kind of techniques, or algorithm, etc, was selected for a procedure.
Response: We have included a Flow diagram to facilite the comprehension of methodology. See figure 1. Line 361.
Information on its physicochemical properties, such as net charge, percentage of hydrophobicity, hydrophobic momentum, and sequence length was obtained from the pep-128 tide sequences found in the transcriptome [45–47]. (How?)
Response: This information was clarified in the lines 133-139.
item 2.3: It is recommended to include the data of synthetic peptides in a table.
Response: Chromatography, mass spectrometry and dichroism data were attached in the supplementary material.
items 2.5 and 2.6: Uniformize the descriptions, sources, and abbreviations of cell lines in the first time they appear in the text.
Response: It was corrected. Line 302, 685 and 687.
Item 2.7: In MTT cytotoxic assay, with the selected cell lines, the peptide concentrations were up to 100 microM. But in the flow cytometry assay, the peptide concentrations were up to 150 microM. And in the apoptosis assay and cell cycle analysis, the peptide concentrations were not specified. It is recommended to uniformize such information.
Response: Because we did a pre-test at those concentrations and they gave us values higher than 33.33 uM. Therefore, we increased the range up to 150 uM to observe more clearly a cytotoxic effect of the peptides.
- Results
In general, the interpretation of "results" should be more concise and less speculative and concentrate the discussion on their own results.
Line 363: This sentence sounds uncomprehensible.
Response: The statement was rewritten. Line 389 -394.
Figure 2, Legend: Is angiotensin a biological activity?
Response: It was corrected to “hormone”. Line 407.
Lines 431-433: This affirmation is contradictory, isn't it?
Response: The statement was rewritten to clarify. Lines 454-457.
Figure 3, Legend: For clarity, it is recommended to include a legend with each procedure used to obtain peptides as depicted in this funnel diagram.
For example, describing the meaning of abbreviature procedures between parenthesis;
Response: The meaning of the abbreviations in the legend were added. Lines 464-466.
Line 442: vNa+-channel from what kind of organism?
Response: The reference articles do not clarify the information, but when aligned to the non-redundant NCBI database, the species of the genus Drosophila appears with 100% identity. The genus was added on line 469.
Line 458: the authors speculated that pugnins might be cell-penetrating. Why this was not experimentally checked along with the other activities? It could be discussed.
Response: For this paper we only focused on testing two activities. In a future paper we will go deeper into this cell penetrating activity.
Figure 4: Symbols are confusing since "+" appears as positive for a given activity.
Response: This was clarified with a legend in the figure. Line 495.
Line 481: "alkaline", is this term used for amino acid residues?
Response: The term was eliminated to avoid confusion.
Lines 523-524: This sentence sounds strange. What kind of solution?
Response: The sentence was rewritten for better understanding. Lines 551-552.
Lines 526-528: Not necessarily this affirmation is true. Some active AMPs are random coils in solution but adopts helicoidal structure upon membrane interaction.
Response: We add reference literature on the change from random coil to membrane helix. Lines 552-554.
Line 613: By taking a look at the data, both peptides are only slightly cytotoxic!
Response: We are agree.
Although the pugnins peptides have an apparent anticancer activity, this only is observable with 100 uM of peptide - a concentration several times higher than those concentrations that cause hemolysis. So, these peptide seems, accordingly to experimental data, to be an antimicrobial peptide, not an efficient anticancer peptide.
Response: these peptides can be the template for the design of better versions.
Line 669: the concentration is relatively high for a promising activity;
Response: We agree, however, that these sequences could be used as templates to improve activities through mutations.
Line 672: Again, why the level of cell death caused by peptides in the annexin-V/SYTOX diverged from the MTT assay?
Response:
MTT is an indirect measure of cell death. It actually measures cellular metabolic activity. MTT is nor suitable to show apoptosis, it is a proliferation or survival assay.The assay is based on the assumption that if you have more dead cells (regardless of the manner of cell death), you have less expression of oxidoreductases and hence less MTT reduction. The thing to note here is that a reduced MTT signal could be reflective of apoptosis, necroptosis or any manner of cell death and sometimes, not even cell death.
Flow cytometry results are more reliable, and you can distinguish the various forms of cell death from each other
Figure 14 and results: If at 150 uM pugnins cause cell death, what is the rationale to check the phase of the cell cycle of dying cells at lethal doses?
Response: The lethality of 150uM was unknown since peptides were evaluated up to 100uM by MTT. In this way the concentration of peptides was raised in the flow cytometry assay in order to see the increased cytotoxicity they could cause in cancer cells.
Discussion:
General: Again, it is recommended to the author to be less speculative and make a correlation with their findings.
Based on the present results, pugnins peptides are antimicrobial that work at 10 -20 uM range, but they are slightly cytotoxic, with a residual cytotoxic (anticancer) activity that is effective with such a high concentration of peptides (over 150 uM).
Lines 743-747: The influence of positively charged amino acid residues and the effect on peptides on bacteria have been studied. It is important to update this information.
Response: It was added two recent papers on the role of positively charged residues in antibacterial peptides. Line 814.
Line 776: In this part of the discussion, authors could correlate the structure-activity of pugnin A and B and the antibacterial selectivity, regarding cell wall and membrane composition.
Ref 134, Line 813: Does this reference mention pugnins? It is recommended to specify what it refers.
Response: A sentence was added to clarify the statement. Line 883-885.
Lines 838-841: For this and the other experimental reasons, pugnins appear as a potential AMP but not a good anticancer peptide.
Response: However, as mentioned above, the sequence can serve as a template for the design and improvement of activities, based on the data obtained.
Conclusion:
This should be re-elaborated and rewritten after the overall amendment of the manuscript to deliver the most relevant findings.
Response: The conclusión was rewritten to deliver the most relevant findings.
Round 2
Reviewer 2 Report
The authors improved almost all the suggestions for publication and proofreading was performed, which significantly improved the paper. I am now satisfied with the concerns and the paper can be published in the present form.
Author Response
Response to Reviewers Comments
April 9th, 2021
Executive Editor
Molecules
Dear Executive Editor,
REVIEWER 2 COMMENTS
We appreciate the reviewer’s comments.
Reviewer 3 Report
The authors answered most of the concerns regarding the data presented in the manuscript.
Only one observation at the end.
As the authors agree and answered the criticism "these peptides can be the template for the design of better versions".
Therefore, in Line 931, I suggested modifying the paragraph like exemplified:
Although the antimicrobial and anticancer activity of this peptide has a variable level of efficacy, the peptide shows a selective antitumor activity towards human prostatic cancer cells (PC3), which would be attractive as a template for further structure-activity improvement and studies
Author Response
Response to Reviewers Comments
April 9th, 2021
Executive Editor
Molecules
Dear Executive Editor,
To facilitate the review of our responses, the following is a point-by-point response to the comments of reviewers. The changes are highlighted in yellow in the text.
REVIEWER 3 COMMENTS
We appreciate the reviewer’s comments.
The authors answered most of the concerns regarding the data presented in the manuscript.
Only one observation at the end.
As the authors agree and answered the criticism "these peptides can be the template for the design of better versions".
Therefore, in Line 931, I suggested modifying the paragraph like exemplified:
Although the antimicrobial and anticancer activity of this peptide has a variable level of efficacy, the peptide shows a selective antitumor activity towards human prostatic cancer cells (PC3), which would be attractive as a template for further structure-activity improvement and studies
Response: Thank you so much for your contribution. We did the changes in Line 931 as you suggested.